# SpecPrune-VLA:

# Accelerating Vision-Language-Action Models via Action-Aware Self-Speculative Pruning

## Abstract

Pruning is a typical acceleration technique for compute-bound models by removing computation on unimportant values. Recently, it has been applied to accelerate Vision-Language-Action (VLA) model inference. However, existing methods focus on local information from the current action step and ignore the global context, leading to >20% success rate drop and limited speedup in some scenarios. In this paper, we point out **spatial-temporal consistency** in VLA tasks: input images in consecutive steps exhibit high similarity, and propose the key insight that token selection should combine local information with global context of the model. Based on this, we propose SpecPrune-VLA, a training-free, two-level pruning method with heuristic control. **(1) Action-level static pruning.** We leverage global history and local attention to statically reduce visual tokens per action. **(2) Layer-level dynamic pruning.** We prune tokens adaptively per layer based on layer-wise importance. **(3) Lightweight action-aware controller:** We classify actions as coarse- or fine-grained by the speed of the end effector. Fine-grained actions are pruning-sensitive, so the controller adjusts pruning aggressiveness accordingly. Extensive experiments show that, compared to the high-performing VLA model OpenVLA-OFT, SpecPrune-VLA achieves up to **1.57×** speedup in the LIBERO simulation benchmark across different hardware configurations and an average speedup of **1.70×** in **real-world robotic tasks** with negligible degradation in task success rate.

## 1 Introduction

Vision-Language-Action (VLA) models, built upon large language models (LLMs), have gained attention for their ability to interpret multimodal inputs and generate robotic actions. Models like RT-1 (Brohan et al., 2022) and OpenVLA Kim et al. (2024) demonstrate strong cross-task generalization and instruction-following capabilities from real-world robot data. Follow-up works Team et al. (2024); Li et al. (2024b); Black et al. (2024); Kim et al. (2025) further are proposed for real-time performance improvement. As shown in Figure 1(a), VLA inference typically involves: (1) tokenizers for encoding multimodal inputs, (2) an LLM backbone for processing multimodal tokens and generating intermediate outputs, and (3) an action head for producing low-level actions. We profile three representative models, OpenVLA Kim et al. (2024), CogACT Li et al. (2024b), and OpenVLA-OFT Kim et al. (2025)in Figure 1(b) and reveal that the LLM is the critical inference bottleneck(*e.g.*, > 70% of the end-to-end latency in even the most efficient model). Therefore, most works target on the LLM acceleration via various acceleration methods Xu et al. (2025b); Hong et al. (2024); Park et al. (2024); Yue et al. (2024); Xu et al. (2025a). However, they largely overlook VLA-specific computation patterns, limiting their effectiveness. Nowadays, the latest VLA models (*e.g.*, CogACT and OpenVLA-OFT) adopt the single-step paradigm that the model directly predicts a sequence of low-level actions through a single LLM forward(*i.e.*, only prefill phase) with hundreds of multimodal tokens. As a result, from the perspective of arithmetic intensity (*i.e.*, the amount of computation per byte) in the hardware roofline model, the VLA inference is primarily compute-bound, shown in Figure 1(c), where latency mainly arises from the computation

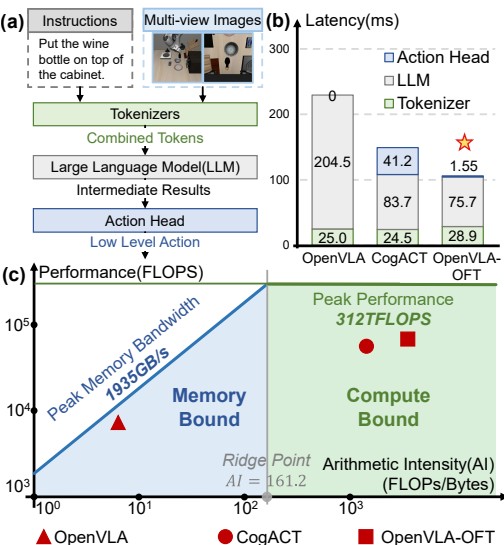

Figure 1: (a) The mainstream inference dataflow of VLA models. (b) Latency breakdown in three typical VLA models in the LIBERO benchmark during each action generation. (c) The practical arithmetic intensity of three models in the roofline model of NVIDIA A800 GPU.

amount rather than memory access.

Pruning is a typical acceleration method for compute-bound problems by effectively reducing the computation Zhou et al. (2024). However, existing token-pruning methods in VLA models Yang et al. (2025); Li et al. (2025) only consider the local information (*e.g.*, the layer results in current action generation) and ignore the global information across the whole model, leading to either > 20% success rate loss or limited speedup in some scenarios.

In this paper, we point out that input images in consecutive action generations exhibit high similarity due to the short temporal intervals between them. Therefore, we consider that the global information from previous inference steps can be leveraged for more reliable and efficient token pruning. Based on the above insight, we propose SpecPrune-VLA, an acceleration method for Vision-Language-Action Models through action-aware self-speculative pruning. The techniques of SpecPrune-VLA can be summarized into three points as follows.

**(1) Action-level static pruning.** Based on the insight, we point out that tokens between consecutive action generation are largely overlapped (*e.g.*, the background in the environment), leading to significant information redundancy. Therefore, we reuse the attention information of the global model(the middle layer and deep layer) from the last generation to prune globally unimportant tokens. Then we enhance it with dynamic elements and current task-relevant tokens by speed-based frame comparison and self-speculative token selection. By fusing tokens selected from local and global levels, we can prune 60% to 70% visual tokens at the beginning of LLM forward.

**(2) Layer-level dynamic pruning.** As the input features propagate through the LLM backbone, the local context of each token is progressively enriched by deeper layers. Therefore, we introduce layer-wise pruning by dynamically updating tokens' importance scores and re-evaluating token importance at different depths. This allows the model to adaptively refine computation focus and remove redundant tokens as contextual understanding matures, reducing extra 20% computation.

**(3) Lightweight action-aware Controller.** We propose that not all actions are equally sensitive to token pruning. Therefore, we categorize actions into coarse-grained (*e.g.*, large translations) and fine-grained (*e.g.*, grasping) types and design a controller. It determines action granularity based on the speed of the end-effector and adaptively adjusts the pruning aggressiveness with negligible overhead, enabling a robust trade-off between speed and accuracy across diverse robotic tasks.

Compared to the high-performing model OpenVLA-OFT, our method achieves an average 1.46× speedup on NVIDIA A800-80GB and 1.57× speedup on NVIDIA GeForce RTX 3090 with negligible degradation in task success rate on the LIBERO simulation benchmark. On the real-world robot, our method exhibits higher potential and achieves 1.70× speedup across various tasks.

## 2 RELATED WORKS

### 2.1 VISION-LANGUAGE-ACTION (VLA) MODELS

VLA models are typically LLM-based Zitkovich et al. (2023); Liu et al. (2023b); Wang et al. (2024), fine-tuned on large-scale simulated Liu et al. (2023a) and real-world O'Neill et al. (2024) robotic datasets. They process multimodal inputs (e.g., images + text) to generate low-level robotic actions. Continuous action spaces are preferred for higher manipulation accuracy Liu et al. (2025), often decoded via lightweight MLPs or diffusion heads Liu et al. (2024); Wen et al. (2025). To ensure high

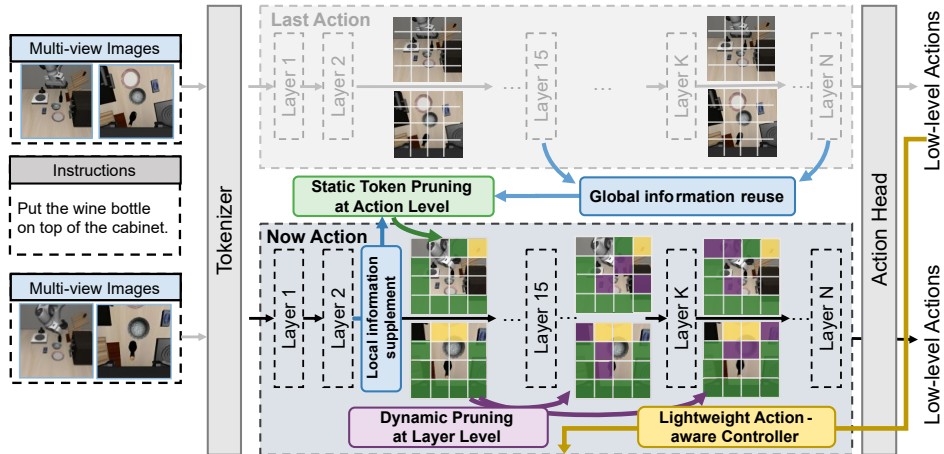

Figure 2: Overview of SpecPrune-VLA. We prune the visual tokens with global and local information with a lightweight action-aware controller.

control frequency and temporal coherence, modern VLAs adopt ACT Zhao et al. (2023), diffusion models Peebles & Xie (2023), or parallel decoding for chunked action generation Li et al. (2024a).

## 2.2 TOKEN-LEVEL ACCELERATION FOR VLA MODEL

Recent works explore token caching or pruning. VLA-Cache Xu et al. (2025b) reuses cached key-value pairs from unimportant tokens, but only reduces 17–25% of total FLOPs and introduces additional GPU memory access overhead. EfficientVLA Yang et al. (2025) prunes visual tokens using single-layer attention heuristics and supplements with diverse patches — yet this risks introducing task-irrelevant content and lacks global context awareness. SP-VLA Li et al. (2025) retains tokens with high vision encoder saliency to preserve spatial-semantic structure, but still fails to filter semantically redundant tokens, leaving unnecessary computation.

## 2.3 SELF-SPECULATIVE DECODING AND LIGHTWEIGHT PREDICTORS

Unlike standard speculative decoding Leviathan et al. (2023) requiring a separate draft model, LayerSkip Elhoushi et al. (2024) uses early layers of the same model for drafting and deeper layers for verification, reducing memory and latency. Separately, SpecEE Xu et al. (2025a) employs a lightweight predictor to shrink the output vocabulary by filtering low-probability tokens, significantly lowering decoding cost.

## 3 KEY INSIGHTS

### 3.1 WHAT REALLY MATTERS IN THE IMAGE

We systematically study which image components are critical for Vision-Language-Action (VLA) models. As shown in Figure 3, Insight 1(a), attention patterns evolve across layers. In the task "put the bowl on the plate", shallow layers attend broadly, including background and irrelevant regions (e.g., table) but miss important object (e.g. bowl and plate); middle layers focus on semantically relevant objects that inform task understanding even though they may not involve with the action (e.g., cabinet); deep layers focus on action-centric tokens directly involved in execution (e.g. plate).

To assess the value of this hierarchical attention, we conduct a post-hoc token pruning experiment. Using attention scores between visual to text ( equation 2)—a commonly used importance proxy Zhang et al. (2024b;a); Ye et al. (2025)—we identify layer-wise important tokens. Actions are first generated without execution, then tokens are pruned based on attention scores, and actions are regenerated from the compressed input and executed.

Results (Figure 3, Insight 1(e)) show that random pruning maintains performance only up to 12.5% sparsity, beyond which accuracy drops sharply, which indicates redundancy exists but also informed pruning guidance is needed. Pruning guided by shallow layers performs poorly under high sparsity

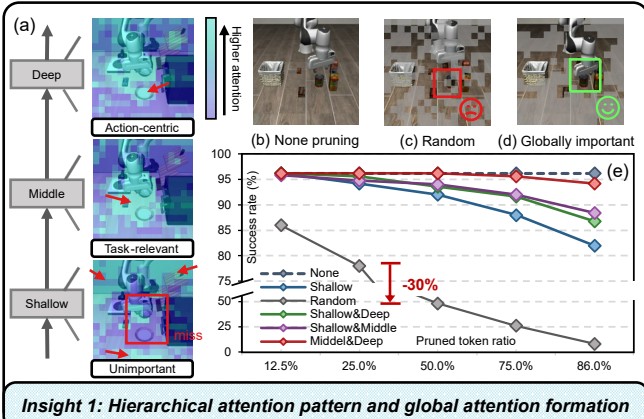 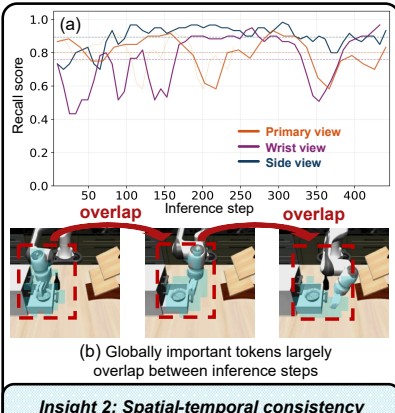

Figure 3: Insight 1: (a) Layers of different depth focus on different information. (b)(c)(d) In pick and place task, random pruning causes important tokens missed out and pruning based on global attention retain important tokens and ensure success rate. (e) The performance of different guiding layer selection. Insight 2: (a) The recall score of Top-30 globally important token sets of consecutive inference remains high in different camera view. (b) The visualization of globally important tokens between consecutive steps.

($>$10% drop), as they capture irrelevant, redundant information (e.g. table texture and background). In contrast, strategies combining mid and deep layers achieve superior robustness, with minimal degradation even at 86% pruning. This demonstrates that fusing task-relevant and action-centric representations provides a reliable signal for effective model compression.

## 3.2 INFORMATION LARGELY OVERLAPS IN IMAGES OF CONSECUTIVE INFERENCE

***Globally important tokens*** needs to be retained to ensure accuracy. It is challenging to identify these tokens before the whole model completes current inference. Current methods, such as Li et al. (2025); Yang et al. (2025), utilize local information such as the attention score of one LLM layer or the vision encoder. However, they didn't consider global information from the whole model and are thus not reliable. In this paper, we emphasize that in VLA models, the overall task goal remains constant and a large proportion of the visual scene remains unchanged across consecutive inferences due to the minimal time change. Therefore, tokens identified as globally important in the previous generation are likely to remain important in the current step as shown in Figure 3, Insight 2(b). We call this spatial-temporal consistency.

To quantify this consistency in token importance, we define the ***Recall of Important Tokens***, which measures the overlap between the globally important token set at the previous step $V_{t-1}$ and the current set $V_t$, normalized by the size of the current set. Formally, it is expressed as:

$$\text{Recall}(V_{t-1}, V_t) = \frac{|V_{t-1} \cap V_t|}{|V_t|}. \tag{1}$$

As shown in Figure 3, Insight 2(a), we observe that this recall reaches an average of 75% -88% from different viewpoints throughout the task execution, indicating strong temporal persistence in token relevance. This temporal consistency inspires us to reuse the global attention scores across time.

## 4 ACTION-LEVEL STATIC TOKEN PRUNING

### 4.1 METHOD

#### 4.1.1 PRUNING BASED ON GLOBAL INFORMATION

As illustrated in Section 3.1, in cross-attention layers where visual tokens serve as queries and text tokens as keys, a high attention weight from a visual token $V_i$ to task-instruction (text) tokens indicates that the visual token is important.

Formally, for a given layer $l$, let $V_i$ be a visual token and $T = \{t_1, t_2, \ldots, t_m\}$ be the text sequence describing the task. We compute the task attention score of $V_i$ as the mean attention it assigns to all text tokens across all heads:

$$\text{Score}_l(V_i) = \frac{1}{H \cdot m} \sum_{h=1}^{H} \sum_{j=1}^{m} A_l^h(V_i, t_j), \tag{2}$$

where $A_l^h(V_i, t_j)$ denotes the attention weight from visual token $V_i$ to text token $t_j$ in attention head $h$ of layer $l$. We define $V_{global}$ as the set of the top-$K_{global}$ visual tokens with the highest such attention scores from the middle and deep layers (we choose the 15th and 32nd layers) in the prior inference step. Based on our key insight that global information exhibits temporal consistency across consecutive actions, we retain $V_{global}$ in the current step.

### 4.1.2 SUPPLEMENTATION OF DYNAMIC TOKENS

Visual tokens undergoing significant changes between inference steps cannot be reliably pruned using global information from the prior step. To preserve up-to-date content, we explicitly retain these *dynamic* tokens during static pruning. Given frames $I_m$ and $I_n$, we partition each into $N \times N$ patches according to the token size. Let $\mathbf{P}_t^{i,j}$ denote the feature vector of patch $(i, j)$ in frame $I_t$. The cosine similarity between corresponding patches is:

$$\text{Sim}(\mathbf{P}_m^{i,j}, \mathbf{P}_n^{i,j}) = \frac{\mathbf{P}_m^{i,j} \cdot \mathbf{P}_n^{i,j}}{\|\mathbf{P}_m^{i,j}\|_2 \|\mathbf{P}_n^{i,j}\|_2}.$$

To identify dynamic tokens, we first filter patches with similarity scores below a threshold $\tau$, then select the top-$k$ patches with the lowest similarity scores from the remaining candidates. Formally, let $\mathcal{P}_n = \{\mathbf{P}_n^{i,j} \mid 1 \le i, j \le N\}$ be the set of all patches in frame $I_n$. We define the candidate dynamic patches as those with significant changes:

$$\mathcal{C}_n = \{\mathbf{P}_n^{i,j} \in \mathcal{P}_n \mid \text{Sim}(\mathbf{P}_m^{i,j}, \mathbf{P}_n^{i,j}) < \tau\}. \tag{3}$$

The most dynamic $K_{dynamic}$ tokens are then given by:

$$V_{dynamic} = \text{Low-}K_{dynamic}\left(\{\text{Sim}_{i,j} \mid \mathbf{P}_t^{i,j} \in \mathcal{C}_t\}\right), \tag{4}$$

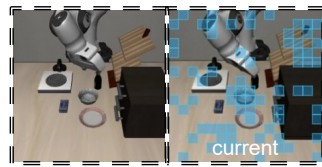

(a) Compare with last consecutive frame

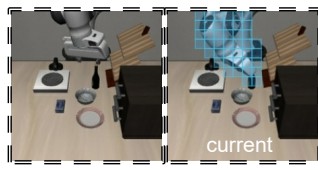

(b) Compare with frame chosen by velocity

Figure 4: Adaptive selection strategy for frame comparison.

Additionally, as shown in Figure 4(a), directly comparing adjacent frames can yield inaccurate results due to camera noise and light changes, especially in real-world scenarios. Therefore, we propose a ***velocity-based frame sampling*** strategy. This method selects a historical reference frame that is $T$ frames before the current one, where $T$ is calculated as: $T = \lfloor b + k \cdot v \rfloor + 4$. Here, $k = -1$ and $b = 7$ are constants based on experimental results. $k$ inversely relates speed $v$ to $T$, while $b$ adjusts the baseline value of $T$. The translational speed $v$ is discussed in Section 6.

### 4.1.3 PRUNING BASED ON LOCAL INFORMATION

Due to changing sub-goals, we need to incorporate information of current generation by analyzing attention-based importance using equation 2. Motivated by works in Section 2.3, we observed that 80%–90% of top-k important tokens from the first two layers reappear in the final layer's top-k (Figure 5II, k=30), indicating early-layer attention provides reliable guidance for token selection. Besides, the first layer alone shows a low hit rate and adding the third layer provides marginal gains with extra latency. Considering precision and efficiency, we use the first two layers for speculation to filter current important tokens.

In each layer, we select the $K_L$ visual tokens with the highest attention scores to form a candidate set $V_{(1)}$ and $V_{(2)}$ respectively, and take the union of these two sets as the local information representation: $V_{local} = V_{(1)} \cup V_{(2)}$. Finally, all the retained token set is: $V_{retain} = V_{global} \cup V_{dynamic} \cup V_{local}$.

## 5 LAYER-LEVEL DYNAMIC TOKEN PRUNING

To preserve the most important tokens in layers, we propose dynamic importance scoring mechanism that leverages attention scores and layer confidence across LLM layers to prune tokens within layers.

### 5.1 IMPORTANCE SCORE FORMULATION

The token importance score is initialized for the remaining visual tokens after static token pruning and subsequently updated in the target transformer layers. The importance score $s_i^{(l)}$ takes into

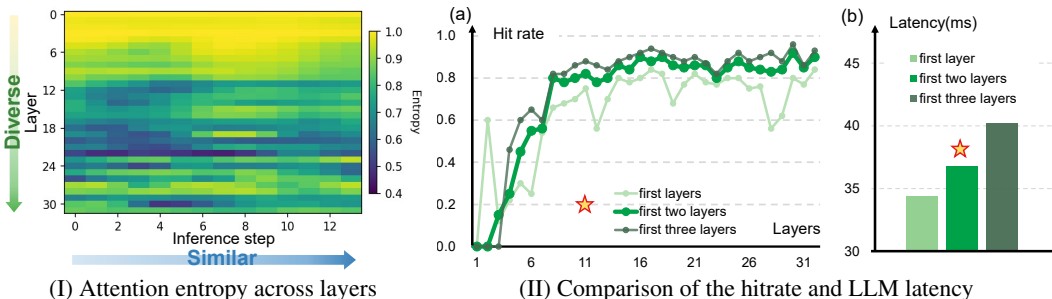

(I) Attention entropy across layers      (II) Comparison of the hitrate and LLM latency

Figure 5: (I) Attention entropy differs across layers, but are similar throughout the task. (II) (a) Comparison of the hitrate between leveraging the first one, two, and three layers. (b) LLM latency comparison between leveraging the first one, two, and three layers.

account both the relative importance weight of tokens and the layer contribution:

$$s_i^{(l)} = \omega_{\text{rank},i}^{(l)} \times \omega_{\text{conf}}^{(l)} \tag{5}$$

where $\omega_{\text{rank},i}^{(l)}$ denotes *rank-based weight* reflecting the token's relative importance in attention ranking, $\omega_{\text{conf}}^{(l)}$ denotes *layer confidence score* measuring the layer's reliability

**Rank-based Weight** For each attention head, visual tokens are ranked based on their image-to-text attention scores in equation 2. To emphasize the contribution of the most important tokens while maintaining a smooth decay in influence, we introduce a rank-based weighting scheme. This weight is defined as:

$$\omega_{\text{rank},i}^{(l)} = \frac{\sigma(-k \cdot \text{rank}_i^{(l)})}{\sum_j \sigma(-k \cdot \text{rank}_j^{(l)})} \tag{6}$$

where $\text{rank}_i^{(l)}$ is the attention ranking of token $t_i$ in layer $l$ and $\sigma(x)$ denotes the sigmoid function, which amplifies the differences between token rankings by mapping them to a smooth range, ensuring that higher-ranked tokens receive significantly more emphasis.

**Layer Confidence Score** Inspired by Zhang et al. (2025), in Transformer layers, high attention entropy indicates a dispersed attention distribution, where the model fails to concentrate on salient tokens. As shown in Figure 5I, we observe that the attention entropy across layers in our VLA model varies significantly with depth, suggesting that different layers contribute unequally to identifying globally important information.

We posit that layers with low entropy, focused attention are more reliable for token importance estimation. Let $A_{ij}^{(l)}$ denote the attention weight from text query token $i$ to image key token $j$ in layer $l$ of the image-to-text attention. The average attention entropy is computed as:

$$\bar{H}^{(l)} = -\frac{1}{N} \sum_{i=1}^{N} \sum_{j=1}^{M} A_{ij}^{(l)} \log A_{ij}^{(l)}, \tag{7}$$

where $N$ and $M$ are the numbers of query and key tokens, respectively. We compute the layer confidence score $\omega_{\text{conf}}^{(l)}$ as:

$$\omega_{\text{conf}}^{(l)} = \frac{1}{\bar{H}^{(l)} + \epsilon}, \tag{8}$$

with $\epsilon > 0$ for numerical stability. Lower entropy corresponds to higher confidence, reflecting more focused and semantically grounded attention. This value is computed in the first inference step and reused thereafter due to high inter-step similarity (see Appendix A.4).

## 5.2 DYNAMIC UPDATING MECHANISM

The final importance score $S_i$ for each token $t_i$ is maintained through an exponential moving average across layers:

$$S_i^{(l)} = (1 - \beta) \cdot S_i^{(l-1)} + \beta \cdot s_i^{(l)} \tag{9}$$

where $\beta$ is the learning rate controlling the update speed, set to 0.2, and $S_i^{(0)} = 0$ is set for initialization. For layers in update layer set, we prune 10% tokens with the lowest score.

# 6 LIGHTWEIGHT ACTION-AWARE CONTROLLER

## 6.1 OBSERVATION AND INSIGHT

Empirically, aggressive token pruning leads to a drop in success rate. Frame-by-frame observation reveals that failures predominantly occurred during object-contact phases, such as manipulation or placement (Figure 6(b)) where even minor errors cause task failure. The task merely fails when those actions are successfully executed. This highlights that task success critically depends on *fine-grained* actions, which demand high precision and are sensitive to pruning. In contrast, *coarse-grained* actions (*e.g.* moving to a general location) tolerate more approximation. Specifically, when the robot approaches an object, fine-grained control is essential for stable contact and successful execution. Thus, action granularity dictates the required level of visual fidelity and inference precision.

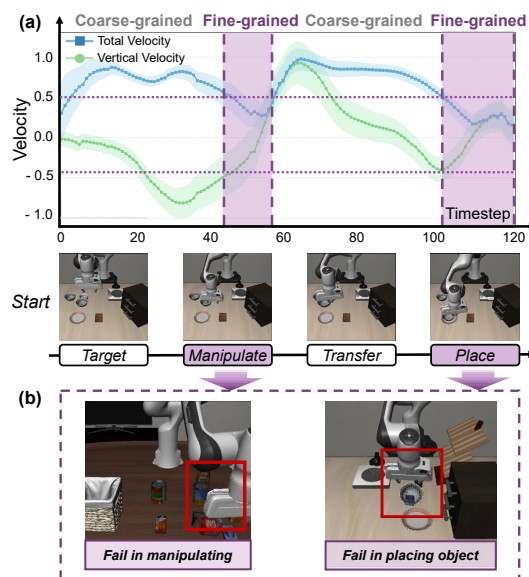

Figure 6: (a) The task process consists of four stages, categorized into coarse- and fine-grained actions based on velocity. (b) Typical failures in fine-grained stages.

Inspired by this, we propose an action-aware pruning strategy: by detecting whether a step requires fine or coarse control, our method preserves more tokens during fine-grained phases and prunes more aggressively during coarse-grained ones, improving both efficiency and success rate.

## 6.2 METHOD

Robotic execution can be divided into four phases that reflects intrinsic task semantics : targeting, manipulating, transferring, and placing (Figure 6(a)). Manipulating and placing require high precision and exhibit naturally low translational and rotational velocities, with non-positive $z$-axis displacement $\Delta z$—a pattern inherent to fine-grained interaction.

Since actions span a fixed duration, end-effector velocity is measured as displacement per step. As all training data are normalized prior to model input, the output displacements $(\Delta x', \Delta y', \Delta z')$ and angular changes $(\Delta \alpha', \Delta \beta', \Delta \gamma')$ are inherently in normalized form. This normalization ensures that velocity magnitudes lie in a consistent range across tasks and platforms, making our method generalizable and independent of specific robot kinematics or environmental scaling.(Other settings result in Appendix A.3) The translational and rotational velocities are computed as:

$$v_t = \sqrt{(\Delta x')^2 + (\Delta y')^2 + (\Delta z')^2}, \quad v_r = \sqrt{(\Delta \alpha')^2 + (\Delta \beta')^2 + (\Delta \gamma')^2}, \tag{10}$$

Analysis of trajectory data in Figure 6 reveals bimodal velocity distributions between coarse and fine-grained phases. From this, we empirically identify thresholds: $v_t^{\text{th}}, v_r^{\text{th}}$, and a small $v_z^{\text{th}} > 0$. The system enters precise mode when $v_t < v_t^{\text{th}}$, $v_r < v_r^{\text{th}}$, and $\Delta z \leq 0$, and exits upon exceeding $v_t^{\text{th}}$ or $v_r^{\text{th}}$ (*e.g.*, during lifting). This adaptive control balances accuracy and efficiency (Alg 1, Appendix).

# 7 EXPERIMENT

## 7.1 EXPERIMENTAL SETTINGS

**Simulation Benchmarks and Platforms** To validate the generalization of our method, we conduct evaluations both on the LIBERO simulation benchmark Liu et al. (2023a) and in the real world. In simulation, we employ four task suites, LIBERO-Spatial, LIBERO-Object, LIBERO-Goal, and LIBERO-Long, to evaluate the model's capabilities in spatial reasoning, object understanding, goal-

| LIBERO benchmark | Success Rate (%) | | | | Avg. Speedup | Avg. SR(%) | FLOPs |
|---|---|---|---|---|---|---|---|
| Method | Spatial | Object | Goal | Long | | | |
| OpenVLA-OFT | 97.6% | 96.5% | 97.9% | 94.5% | 1.00× | 96.6% | 100% |
| SparseVLM | 96.8% | 94.2% | 97.6% | 93.6% | 1.28× | 95.6% | 77% |
| FastV | 94.6% | 95.8% | 94.0% | 88.8% | 1.44× | 93.3% | 57% |
| DivPrune | 92.4% | 91.2% | 89.0% | 84.8% | 1.46× | 89.4% | 54% |
| VLA-Cache | 99.0% | 97.7% | 97.4% | 93.6% | 1.07× | 96.9% | 83% |
| EfficientVLA | 96.5% | 91.1% | 96.0% | 72.1% | 1.52× | 88.9% | 35% |
| **Ours**(prune rate=0.8) | 97.4% | 95.8% | 97.7% | 93.4% | 1.46× | 96.1% | 43% |

Table 1: Performance Evaluation (Success Rate and Average Speedup)

directed planning and execution, and long-horizon task completion, respectively. All main experiments are conducted on a Linux workstation with an NVIDIA A800-80GB GPU.

**Implementation Details** We select the OpenVLA-OFT Kim et al. (2025) as the code base for the implementation of our techniques. We report end-to-end latency, average speedup and success rate in our result. The latency and speedup may fluctuate slightly across tasks due to the changing environment and our method's adaptability, we report the detail in the Appendix. We repeat every task of every task suite for 40 to 50 times to mitigate the impact of random errors. The latency here is defined as the time duration from when the model receives the input to when it generates the action. We report the average inference time over the ten tasks within each task suite.

**Baselines** We select OpenVLA-OFT and $\pi_0$ as our target model($\pi_0$ result is in Appendix). It utilizes DINOv2 Oquab et al. (2023) and SigLIP Zhai et al. (2023) as visual encoders to extract visual features, Llama2-7B as the backbone LLM, and a four-layer MLP as an action head to generate continuous actions. The model receives two-view images: the third-person view and the wrist view. We also consider five optimization methods: SparseVLM Zhang et al. (2024b), a framework that adaptively sparsifies less important visual tokens and recycles their information to minimize performance loss, DivPrune Alvar et al. (2025) selects diverse visual tokens to preserve information and maintain accuracy. FastV Chen et al. (2024) prunes redundant visual tokens early in the model based on observed sparse attention patterns for efficient inference. EfficientVLA Yang et al. (2025), a visual pruning approach for VLA models, and VLA-Cache Xu et al. (2025b) , which leverages image similarity to cache features across time steps.

## 7.2 PARAMETER SETUP

Our base K values are chosen to maximize the temporal consistency of important information, measured by the ***Recall of Important Tokens***( equation 1) as mentioned in our motivation. For the $K_{global}$, we evaluated value of 20, 30, 40 and observed the average recall rate Recall$(V_{t-1}, V_t) = |V_{t-1} \cap V_t|/|V_t|$ is the highest when $K_{global}$ equals 30. For $K_{local}$ and $K_{dynamic}$, we tested several pairs: (12, 15), (24, 20), (36, 30). The token set $\hat{V}_t$ ($V_t$ supplemented by local and dynamic tokens), reaches the highest recall rate Recall$(V_{t-1}, \hat{V}_t) = |V_{t-1} \cap \hat{V}_t|/|\hat{V}_t|$ when the pair equals (24, 20). Therefore, we set the base value $K_{global}$=30, $K_{local}$=24 and $K_{dynamic}$=20.

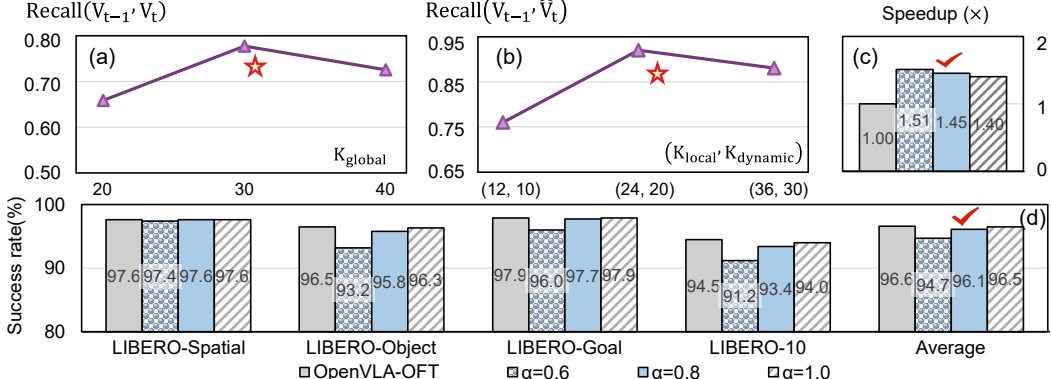

Figure 7: Ablation on base K value and prune rate

## 7.3 DESIGN SPACE EXPLORATION

We use the prune ratio $\alpha$ to adjust the overall K values by scaling $K_i = \alpha \cdot K_i$ (The detail setup is in A.6.2) to control the aggressiveness of our pruning. Therefore we conduct a design space exploration to explore the impact of different prune ratios. The smaller the prune ratio, the more tokens are pruned, leading to a drop in success rate and a rise in speedup. To balance accuracy and speed, we set the prune ratio to 0.8 for universal setup.

## 7.4 EVALUATION ON SPEEDUP AND SUCCESS RATE

Table 1 shows the end-to-end evaluation on success rate (SR), latency and speedup on four LIBERO task suits. SpecPrune-VLA reduces FLOPs by 57% and achieves an average speedup of $1.45\times$ with negligible loss ($< 0.5\%$) in success rate. SparseVLM yields limited speedup and degraded SR, as its pruning, recycling, and merging strategy only minimally reduces computation and is not suited for precise action generation. Compared to FastV and DivPrune (both retaining similar token counts with our method for fairness), our method achieves comparable speedup but superior SR. FastV prunes tokens using early-layer attention without training, ignoring global context; DivPrune maximizes feature diversity but neglects task-relevant token importance in VLA models. VLA-Cache maintains high SR by mitigating noise and improving motion continuity through caching, yet achieves limited speedup due to small computation reduction (17%) and GPU memory access overhead. EfficientVLA (L=28, T=112) attains higher speedup by skipping layers and aggressive token pruning, but suffers notable SR drops in certain scenarios by compromising critical action-related information in hidden states—this approach was originally designed for VLA models with a diffusion-based action expert, which may better tolerate perturbations in action outputs.

## 7.5 ABLATION STUDY

**Ablation on global attention reuse**   We compare recall rate as  equation 1 and success rate of pruning with and without global attention reuse in static pruning stage. The recall rate represents the overlap between final retain token and ground-truth important token set. For fair comparison, we set $K_{\text{local}}$= 54 when without global attention reuse, otherwise, we set $K_{\text{local}} = 24$, $K_{\text{global}} = 30$. As shown in Table 2, incorporating global context yields better token recall and higher success rates, indicating that prior-step attention provides valuable, task-relevant and action-centric guidance. This supports our key insight: spatio-temporal coherence in robotic tasks makes previous-step global attention a reliable prior for the current inference step.

**Ablation on entropy-based layer weight**   We compare recall rate and success rate of pruning with and without entropy-based layer weighting in dynamic pruning stage in Table 3. From the perspective of recall rate, average weighting utilizes uncertain information from high-entropy layers and mistakenly prunes globally important tokens, thus achieving poorer success rate.

| Method | Recall (%) | LIBERO SR (%) |
|---|---|---|
| w/ global attn. | 92% | 96.1% |
| w/o global attn. | 84% | 93.4% |

Table 2: Ablation on global attention ablation

| Method | Recall (%) | LIBERO SR (%) |
|---|---|---|
| Entropy-based | 88% | 96.1% |
| Average | 66% | 92.0% |

Table 3: Ablation on entropy-based layer weight

**Ablation on three techniques**   To evaluate the effectiveness of our proposed method, we conducted an ablation study on the LIBERO-Spatial task suit (Figure 8(a)). Our full model achieves a success rate(SR) of 97.4%, comparable to the baseline (97.6%), and it significantly reduces latency from 109ms to 72.3ms, resulting in a speedup of $1.51\times$ compared to OpenVLA-OFT. This demonstrates the efficiency gains of our approach while maintaining competitive accuracy. The ablation study further highlights the importance of each component: Static (Tech 1) and Dynamic (Tech 2) pruning slightly affect the SR (96.8%) but reduce latency to 70.8ms, indicating that pruning contributes to the overall latency reduction. The introduction of the action-aware controller increases the success rate and causes negligible latency (1.5ms). This suggests that the controller plays a crucial role in maintaining high accuracy.

## 7.6 EXTENDED EVALUATION

**Evaluation on Various Computing Platforms**   To validate the applicability of our method on different devices, we conduct experiments on NVIDIA GeForce RTX 3090. As illustrated in Fig-

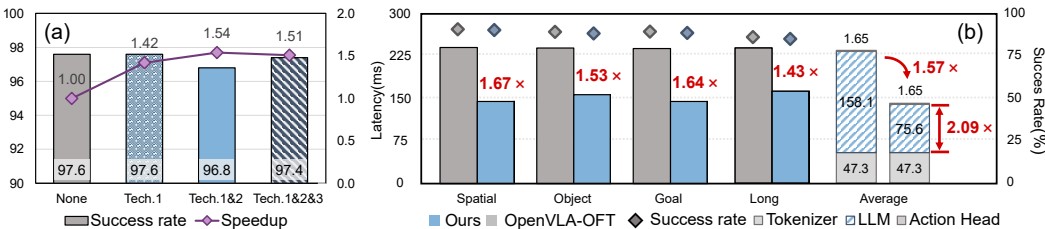

Figure 8: (a) Ablation study in LIBERO-spatial. (b) Extended evaluation on NVIDIA 3090 GPU

ure 8(b), our method achieves an average speedup of **2.09×** in LLM inference time and **1.57×** in end-to-end latency. The results consistently demonstrate improved inference efficiency, underscoring the scalability and effectiveness of our approach under diverse computational conditions.

**Generalization and Robustness analysis**   We conduct experiments on LIBERO-plus Fei et al. (2025) with prune ratio ($\alpha$) = 0.8 to validate the generalization and robustness. LIBERO-plus includes far more and challenging tasks. We choose 200 different long-horizon and object-recognition tasks that adding perturbations to camera viewpoint, light condition, background and noise. With degradation in success smaller than 0.6%, the results confirm that our configuration generalizes well to unseen conditions and is robust to perturbations.

| Method | Camera | Light | Background | Noise | Avg. SR (%) |
|---|---|---|---|---|---|
| OpenVLA-OFT | 96.0% | 88.5% | 95.0% | 97.0% | 94.1% |
| Ours ($\alpha$=0.8) | 94.0% | 87.8% | 95.0% | 97.0% | 93.5% |

Figure 9: Generalization and robustnessperformance on LIBERO-plus under various perturbations.

### 7.7 EVALUATION ON REAL ROBOT

**Experimantal settings**   In this section, we evaluate the real-world performance of our method. We use a Flexiv Rizon4 arm with three cameras with different viewpoints as the Figure 11 shows. We finetune the model via LoRA  Hu et al. (2022) with our collected demonstration data following the configuration in the OpenVLA-OFT training process. More configuration and training details are in the Appendix.

**Results**   Table 4 reports the performance of SpecPrune-VLA in real-world tasks.   Our method achieves a 1.70× speedup while maintaining the success rate. The results show that SpecPrune-VLA is a highly potential acceleration method for VLA models.

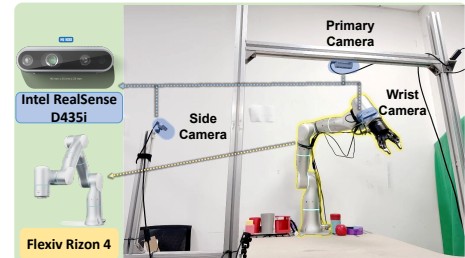

Figure 10: LIBERO-plus

Figure 11: Our real world robot: A Flexiv Rizon4 arm equipped with a gripper and three Intel cameras.   The cameras are mounted separately at three locations: on the wrist, on its side, and above its head.

| Method | Success Rate (%) | | | | Avg Latency (ms) | Avg Speedup |
|---|---|---|---|---|---|---|
| | Pick&Place | TransferTube | PickUpCup | MultiCubeTask | | |
| OpenVLA-OFT | 96.7 | 85.0 | 91.7 | 95.0 | 187.7 | 1.00× |
| Ours | 96.7 | 82.0 | 90.0 | 95.0 | 110.2 | 1.70× |

Table 4: Performance comparison on real-world robot tasks.

## 8 CONCLUSION

We propose SpecPrune-VLA, a training-free, two-level token pruning method that combines local and global information for efficient token selection. On the LIBERO benchmark, it achieves 1.46× and 1.57× speedups on NVIDIA A800 and RTX 3090 GPUs respectively, with negligible success rate drop. In real-world tasks, it delivers a 1.70× speedup without compromising success rate, demonstrating robust cross-hardware performance and strong real-world generalization.

## REPRODUCIBILITY STATEMENT

To ensure reproducibility, we provide full details of hyperparameter settings, experimental environments (including benchmarks, hardware, and training procedures) in the main experiment section and in the appendix in the experiment part.

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

# A APPENDIX

## A.1 USE OF LLM

We used a large language model (LLM) solely for language editing purposes, including grammar correction, sentence restructuring, and clarity improvement. All technical content, methodology, experiments, and conclusions were developed and validated entirely by the authors.

## A.2 THE OVEREALL ALGORITHM OF ACTION-AWARE PRUNING

To clearly illustrate the control flow of our action-aware controller and pruning method, we present in Algorithm 1. For each layer of the LLM, the input is the visual tokens from the previous layer (or initial input), and the output is the pruned set of visual tokens.

If the current layer is among the first two layers, we select the top-$K_{local}$ tokens according to their attention scores and add them to the local important token set. After the second layer, we initialize the retained token set $V_{\text{retain}}$ with the top-$K_{global}$ globally important tokens inferred from the previous action step, and then augment it with locally inferred important tokens (top-$K_{\text{base}}$ from Layer 1 and 2) and $K_{dynamic}$ dynamically selected tokens based on low frame similarity. This constitutes the **static pruning at the action level**.

Subsequently, for each layer, we implement **dynamic pruning at layer level**. We dynamically update token importance scores $s_i^{(l)}$ based on layer-wise confidence and intra-layer attention ranking. At designated pruning layers $L_{\text{prune}}$(see A.4), we perform a further pruning step, retaining only the top $\alpha \times \gamma\%$ tokens with the highest cumulative importance scores $S_i^{(l)}$.

The two-level pruning is scheduled by the action-aware controller. It determines the current action mode(fine or coarse grained) and adaptively adjusts the retain token from current local information.

---

**Algorithm 1** Action-Aware Pruning in One Layer

**Input**: Full token set $V$ (visual tokens from previous layer or initial input)
**Parameters**: Pruning ratio $\alpha$, Velocity thresholds $v_t^{th}, v_r^{th}$, Decay factor $\beta$, Top-K sizes $K_{local}, K_{dynamic}, K_{global}$ and $\gamma\%$, Layer sets $L_{prune}$
**Output**: Retained token set $V_{retain}$

  1: **Action-aware Controller**
  2: Compute translational velocity $v_t$ and rotational velocity $v_r$ from action history
  3: **if** $v_t < v_t^{th}$ **and** $v_r < v_r^{th}$ **and** $\Delta z \le 0$ **then**
  4:     Enter **fine-grained**
  5: **else**
  6:     Enter **coarse-grained**
  7: **end if**
  8: **Static Token Pruning at Action Level**
  9: $V_{global} \leftarrow$ top-$K_{global}$ tokens based on attention scores from the previous action step
 10: Compute frame similarity $Sim(P_t, P_{t-T})$ {Adaptive temporal window based on velocity}
 11: $V_{dynamic} \leftarrow$ lowest $K_{dynamic}$ similarity patches
 12: For layers $l = 1$ and $l = 2$, compute attention scores:
 13:     $V_{local}^{(l)} \leftarrow$ top-$K_{local}$ tokens by attention score in layer $l$
 14: $V_{local} \leftarrow V_{local}^{(1)} \cup V_{local}^{(2)}$
 15: $V_{retain} \leftarrow V_{global} \cup V_{dynamic} \cup V_{local}$ {Initial retained set}
 16: **Dynamic Token Pruning at Layer Level**
 17: **if** current layer $l \notin L_{prune}$ **then**
 18:     $s_i^{(l)} \leftarrow \omega_{rank,i}^{(l)} \times \omega_{conf}^{(l)}$ {Token importance score}
 19:     $S_i^{(l)} \leftarrow (1 - \beta) \cdot S_i^{(l-1)} + \beta \cdot s_i^{(l)}$ {Exponential moving average}
 20: **else**
 21:     $V_{retain} \leftarrow$ tokens with top $\alpha \times \gamma\%$ highest $S_i^{(l)}$
 22: **end if**
 23: **return** $V_{retain}$

---

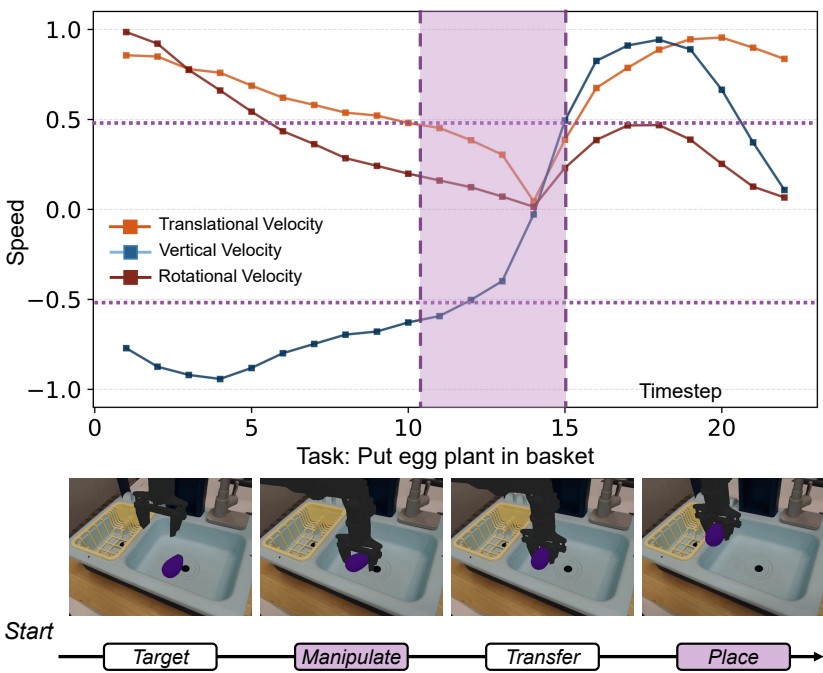

Figure 12: The task process is categorized into coarse- and fine-grained actions based on velocity.

### A.3 GENERALIZATION OF ACTION MODE RECOGNITION

In Section 6.2, we observe that the execution process can be segmented into fine-grained and coarse-grained modes based on thresholds for translational, vertical, and rotational velocities. Our results in both the LIBERO simulation (using a Franka Emika Panda 6-DoF arm) and real-world experiments (using a Flexiv Rizon 4 7-DoF arm) demonstrate the generalization of this method. Furthermore, we show that this characteristic is independent of the specific model, robot, or environment. To validate this, we conduct experiments using a different VLA model, UniVLA Bu et al. (2025) within the SimplerEnv Li et al. (2024c) simulation framework. Notably, SimplerEnv employs a WidowX-250 6-DoF robotic arm, which differs significantly in kinematics and control characteristics from the previous platforms. As shown in the speed profile in Figure 12, the chosen thresholds consistently distinguish between fine-grained and coarse-grained actions across different setups.

### A.4 COMPLEXITY ANALYSIS

Consider a Transformer model with $N$ layers. The computational cost (in FLOPs) per layer when processing $L$ tokens is approximately:
$$\text{FLOPs}_{\text{layer}}(L) = 4LD^2 + 2L^2D + 2LDM$$
where:

- $L$: sequence length (number of tokens)

- $D$: hidden dimension

- $M$: number of attention heads

In typical configurations, $D \gg L$ and $D \gg M$, making the term $4LD^2$ the dominant component, $\text{FLOPs}_{\text{layer}}(L) = 4LD^2 + 2L^2D + 2LDM \approx 4LD^2$. Thus, the computation is primarily driven by the feed-forward and attention projections in the hidden layers, and the overall complexity scales linearly with both sequence length.

**Static Token Pruning** The static token pruning strategy reduces an average of 360 tokens from the original about 600 tokens(*i.e.* 60% sparsity in input), let $L_r = 0.4 \cdot L$ denote the number of retained tokens.

| Prune Ratio $\alpha$ | 0.6 | 0.8 | 1.0 |
|---|---|---|---|
| Pruned Tokens | 382 | 360 | 336 |
| Visual Retention | 25.4% | 29.8% | 34.4% |

Table 5: Number of pruned tokens and visual retention rate under different prune ratios.

Thus, for layers with $L_r$ length input, the FLOPs become $\text{FLOPs}(L_r)$. For a model with H layers, $\text{FLOPs}_{\text{static}} = 2\text{FLOPs}(L) + (H - 2)\text{FLOPs}(L_r)$

**Dynamic Token Pruning** We apply progressive token pruning in the depth interval $[10, 25]$, with pruning layers selected at regular intervals of $T$. The set of pruning layers is defined as:
$$\mathcal{S} = \{s_k = 10 + (k - 1)T \mid s_k \le 25, k = 1, 2, \ldots\},$$
At each pruning layer $s_k$, we reduce the token count by a retention factor $\gamma = 0.9$. Starting from an initial retained length $L_r$, after the $k$-th pruning step (*i.e.*, at layer $s_k$), the token count becomes $\gamma^k L_r$. The pruning interval is T layers.

The total computational cost is then:

$$\text{FLOPs}_{\text{final}} = \underbrace{2 \cdot \text{FLOPs}(L)}_{\text{early layers}} + \underbrace{8 \cdot \text{FLOPs}(L_r)}_{\text{shallow layers}} + \underbrace{\sum_{k=1}^{|\mathcal{S}|-1} T \cdot \text{FLOPs}(\gamma^k L_r)}_{\text{dynamic pruning}}$$
$$+ \underbrace{(H - 10 - (|\mathcal{S}| - 1)T) \cdot \text{FLOPs}(\gamma^{|\mathcal{S}|} L_r)}_{\text{late layers}}, \tag{11}$$

where $H$ is the total number of layers, and $L_r$ is the token count after static pruning.

**Overall FLOPs reduction** In our configuration, the model has 32 layers and T is set to 5. Therefore, $\text{FLOPs}_{\text{static}} = 0.44\text{FLOPs}(L)$, $\text{FLOPs}_{\text{final}} = 0.85\text{FLOPs}_{\text{static}} = 0.37\text{FLOPs}(L)$. Dynamic pruning leads 15% decrease in overall token-level computation across the model. The overall FLOPs reduction in LLM module is 63%.

COMPUTATION OVERHEAD

**Patch similarity:** Calculating patch similarity introduces extra computations. The cosine similarity is calculated based on the raw patches before the image being encoded. The similarity between corresponding patches in two frames is computed as $\text{Sim}(P_m^{i,j}, P_n^{i,j}) = \dfrac{P_m^{i,j} \cdot P_n^{i,j}}{\|P_m^{i,j}\|_2 \|P_n^{i,j}\|_2}$ , For N patches and patch size p ,this operation has complexity $\mathcal{O}(N \cdot p^2)$. Here N=256, p=14.

**Attention entropy:** As defined in equation 7, the attention entropy is computed over the $L \times L$ attention matrix at layer $l$. Its complexity is $\mathcal{O}(L^2)$ and introduces 1ms latency. Naively computing it at every step would incur 32ms latency. To avoid this, we observe that entropy patterns are stable across inference steps within the same task in Figure 5I. Therefore, we compute the layer confidence scores only once during the first inference and reuse them in subsequent steps.

Breakdown of every module shows that the computation overhead (calculating patch similarity calculation and attention entropy) only introduces about 4.5% latency on average, which is negligible.

A.5 FURTHER ANALYSIS ON DIFFERENT VIEW OF IMAGES

Different camera viewpoints present distinct characteristics. In this section, we provide a systematic exposition of the first key insight.

| Latency Breakdown | | | | |
|---|---|---|---|---|
| **Similarity calculation** | **First 2 layers** | **Attention Entropy** | **Our Method** | **Original Model** |
| $1.8 \pm 0.1\,\text{ms}$ | $5.5 \pm 0.2\,\text{ms}$ | $1.0 \pm 0.2\,\text{ms}$ | $72$–$78\,\text{ms}$ | $109.0\,\text{ms}$ |

Table 6: Latency breakdown per module and comparison with the original model.

**In the fixed view(*e.g.* third-person)**, such as high camera and side camera, the robot arm and the objects it touches are the dynamic components, while the task-related patches (*e.g.* objects on the table) are relatively static, as Figure 13 shows. Therefore, the key is to extract the intersection of both — the regions that involve the dynamic and the task-relevant pixels.

**In the dynamic view(*e.g.* wrist-mounted camera)**, although all objects are in motion, the pixel patches along object boundaries exhibit more significant changes. We process and interpret the temporal signal of a pixel patch using the Fourier transform:

$$P(f) = \int_{-\infty}^{\infty} p(t)e^{-j2\pi ft}dt \tag{12}$$

Here $p(t)$ represents pixel intensity at time $t$ , $P(f)$ denotes the frequency-domain representation of the signal, $f$ stands for frequency. Since objects have different colors from the background, the boundary patches will change color at a certain time. Therefore, in these regions, the resulting signal $p(t)$ contains more abrupt changes, which correspond to higher energy in the high-frequency components of $P(f)$. This indicates that boundary patches carry richer temporal dynamics. As a result, task-related patches and dynamic patches can be complementary to some extent, and their intersection can also be used to identify the most critical regions.

## A.6 EXPERIMENT DETAILS

### A.6.1 IMPLEMENTATION OF COMPARATIVE METHODS

We provide details on the implementation of the baseline and comparative methods. All models were implemented on one A800 GPU.

**OpenVLA-OFT**  The latency and the success rate on four datasets are similar to the data in the original paper.

**SparseVLM**  SparseVLM is a visual pruning method for Vision-Language models. It dynamically prunes redundant image tokens layer-wise by leveraging self-attention mechanisms and textual guidance.

**FastV**  FastV prunes visual tokens inVLMs by learning adaptive attention patterns in early layers and removing less attended tokens in subsequent layers, using the LLM's signal to guide pruning in a plug-and-play manner.

**DivPrune**  DivPrune formulates token pruning as a Max-Min Diversity Problem (MMDP), selecting a subset of visual tokens that maximizes diversity to reduce redundancy, enabling effective performance at high pruning ratios without fine-tuning or calibration.

**VLA-Cache**  While the original method was developed on the OpenVLA model, its authors adapted and extended it to the OpenVLA-OFT. All results reported in our experiments are obtained using the authors' official implementation to ensure reproducibility.

**EfficientVLA**  The method focuses on VLA models with diffusion action expert and it also optimizes the action expert. Besides, it has not been open-sourced. As a result, we only re-implement it according to the details of visual token pruning and layer pruning provided in the original paper. Following the reported setup, we retain 28 LLM layers and 112 visual tokens (total of two images) throughout inference.

### A.6.2 PARAMETER SETUP

**Base values for top-Ks** In static token pruning stage, the base value for top-Ks are $K_{global}$=30, $K_{local}$=24 and $K_{dynamic}$=20 according to Section 7.2. The prune ratio $\alpha$ adjust the overall K values by scaling $K_i = \alpha \cdot K_i$. In fine-grained mode, $K_{global}$=30 and $K_{local}$=40 which are higher because the model needs more information to generate fine-grained action. $K_{dynamic}$=10, which is lower because, in fine-grained mode, the speed is much lower, resulting in fewer dynamic tokens will appear.

$$K_{global} = \alpha \times 30$$

$$K_{local} = \alpha \times \begin{cases} 24, & \text{coarse-grained mode,} \\ 40, & \text{fine-grained mode.} \end{cases}$$

$$K_{dynamic} = \alpha \times \begin{cases} 20, & \text{coarse-grained mode,} \\ 10, & \text{fine-grained mode.} \end{cases}$$

**The threshold $\tau$ for Dynamic Token Selection** The threshold $\tau$ for Dynamic Token Selection filters patches with low change magnitude to avoid false positives caused by lighting or camera noise. Since we select the top-$K_{dynamic}$ most dissimilar patches among those below $\tau$, the exact value is not highly sensitive. In simulation, we set $\tau$=0.95 ; in real-world settings with higher noise, we use $\tau$=0.8 . Experiments shows performance remains stable within a range of $\tau \in$[0.9,0.99] in simulation and $\tau \in$[0.6,0.9] in real world, confirming robustness.

**Update Rate $\beta$ in dynamic pruning** Empirically, in exponential moving average (EMA) formulations of the form $S_i^{(l)} = (1-\beta) \cdot S_i^{(l-1)} + \beta \cdot s_i^{(l)}$ , update rate is often set to a small value to ensure smooth calculation. In our method, $\beta$ can be set in a loose range (0.1 - 0.3) without performance loss. However, large value($> 0.6$) will cause loss in task success rate. This is because overly aggressive updates amplify noise from early layers, causing unimportant tokens to be incorrectly assigned high importance scores, while critical tokens may be mistakenly pruned.

**Velocity threshold distinguishing action mode** Based on empirical data from Figure 12, 6, we set the final thresholds as $v_t^{\text{th}} = 0.5$ and $v_r^{\text{th}} = 0.2$. When $v_t < v_t^{\text{th}}$, $v_r < v_r^{\text{th}}$, and $\Delta z \leq 0$, the fine-grained action mode is triggered. An ablation study on the threshold values is conducted on the LIBERO-spatial dataset, as summarized in Table 7.

| $v_t^{\text{th}}$ | 0.3 | 0.5 | 0.7 | | $v_r^{\text{th}}$ | 0.1 | 0.2 | 0.4 |
|---|---|---|---|---|---|---|---|---|
| Success Rate | 96.8 % | 97.4 % | 97.6 % | | Success Rate | 96.8 % | 97.4 % | 97.4 % |

Table 7: Ablation study of success rate under different threshold settings on LIBERO-spatial.

The results show that when the threshold values are too small, the majority of actions are classified as "coarse-grained," leading to aggressive pruning, which causes a noticeable drop in performance (e.g., success rate drop by 0.6%). Conversely, when the threshold values are too large, nearly the entire process is classified as "fine-grained," resulting in more tokens being retained. Although this yields minimal improvement in success rate, it introduces additional latency. Therefore, the chosen thresholds ($v_t^{\text{th}} = 0.5$, $v_r^{\text{th}} = 0.2$) achieve an optimal balance between accuracy and efficiency, ensuring high task success while maximizing inference acceleration.

### A.6.3 DETAIL RESULTS

The detailed latency and speedup of baseline and different methods are listed in Table 8.

The results of different prune ratio are listed in Table 11.

We conducted multiple independent experiments using different random seeds to comprehensively evaluate the performance stability of both the original model and our method. The results are reported as mean success rates with fluctuation ranges, reflecting performance consistency across runs. The data show that, despite minor variations, our method consistently achieves success rates very close to those of the original model. In several cases, our approach demonstrates improved stability. This indicates that our method maintains strong performance while achieving the intended improvements, demonstrating its effectiveness and robustness. Values are reported as mean $\pm$ half-range (*i.e.*, $\pm \frac{\max - \min}{2}$) over multiple runs.

| Method | Latency (ms) / Speedup | | | | Avg. Latency(ms) | Avg. Speedup |
|---|---|---|---|---|---|---|
| | Spatial | Object | Goal | Long | | |
| OpenVLA-OFT | 109.0 / 1.00× | 109.0 / 1.00× | 109.0 / 1.00× | 109.0 / 1.00× | 109.0 | 1.00× |
| SparseVLM | 85.3 / 1.28× | 85.3 / 1.28× | 85.3 / 1.28× | 85.3 / 1.28× | 85.3 | 1.28× |
| VLA-Cache | 101.8 / 1.07× | 101.8 / 1.07× | 101.8 / 1.07× | 101.8 / 1.07× | 101.8 | 1.07× |
| EfficientVLA | 71.7 / 1.52× | 71.7 / 1.52× | 71.7 / 1.52× | 71.7 / 1.52× | 71.7 | 1.52× |
| **Ours($\alpha$= 0.8)** | 74.8 / 1.46× | 74.5 / 1.46× | 74.8 / 1.46× | 75.8 / 1.44× | 75.1 | 1.46× |

Table 8: Performance Evaluation (Latency and Speedup)

| Prune Ratio $\alpha$ | Success Rate (%) / Speedup | | | | Avg. SR (%) | Avg. Speedup |
|---|---|---|---|---|---|---|
| | Spatial | Object | Goal | 10 | | |
| None (OpenVLA-OFT) | 97.6 / 1.00× | 96.5 / 1.00× | 97.9 / 1.00× | 94.5 / 1.00× | 96.6 | 1.00× |
| $\alpha$=1.0 | 97.6 / 1.41× | 96.3 / 1.43× | 97.9 / 1.40× | 94.0 / 1.38× | 96.5 | 1.40× |
| $\alpha$=0.8 | **97.6 / 1.46×** | **95.8 / 1.46×** | **97.7 / 1.46 ×** | **93.4 / 1.44×** | **96.1** | **1.46×** |
| $\alpha$=0.6 | 97.4 / 1.52× | 93.2 / 1.50× | 96.0 / 1.52× | 92.2 / 1.51× | 94.7 | 1.51× |

Table 9: Ablation study on prune rate $\alpha$. Success rates (%) and speedup factors are reported per task. Our method achieves optimal trade-off at $\alpha = 0.8$, achieving $1.45\times$ average speedup with minimal accuracy drop.

**Result on more models** We also implement our method on $\pi_0$ Black et al. (2024), a Transformer based VLA model with a flow-matching action expert. Our method achieves $1.31\times$ speedup with only 0.7% degradation in success rate. This result consistently demonstrates the generalization capability of our method across different model architectures. The speedup is slightly lower compared to OpenVLA-OFT. Because the flow-matching action expert contains more parameters and requires multiple denoising iterations during inference, resulting in a higher proportion of end-to-end latency. Consequently, acceleration of the LLM backbone has a relatively smaller impact on the overall end-to-end performance. In the future, exploration on optimizing the flow-matching module to improve its efficiency can be made.

| Method | Success Rate (%) | | | | Avg. SR (%) |
|---|---|---|---|---|---|
| | LIBERO-Spatial | LIBERO-Object | LIBERO-Goal | LIBERO-10 | |
| OpenVLA-OFT | $98.2 \pm 0.6$ | $96.9 \pm 0.35$ | $97.2 \pm 0.4$ | $94.8 \pm 0.25$ | 96.8 |
| Ours | $98.0 \pm 0.5$ | $96.1 \pm 0.7$ | $97.0 \pm 0.6$ | $94.3 \pm 0.25$ | 96.4 |

Table 10: Average success rates (%) with fluctuation ranges across datasets for the original model and our method.

| Method | Success Rate (%) / Speedup | | | | Avg. Speedup | Avg. SR (%) |
|---|---|---|---|---|---|---|
| | LIBERO-Spatial | LIBERO-Object | LIBERO-Goal | LIBERO-10 | | |
| $\pi_0$ | 96.8 / 1.00× | 98.8 / 1.00× | 95.8 / 1.00× | 85.2 / 1.00× | 1.00× | 94.2 |
| $\pi_0$+Ours | 96.6 / 1.32× | 98.0 / 1.31× | 95.2 / 1.32× | 84.2 / 1.30× | 1.31× | 93.5 |

Table 11: Performance evaluation on $\pi_0$ model.

### A.6.4 EXPERIMENT VISUALIZATION

To understand what the model actually observed when completing a task, we visualize the retained visual tokens. The colored patches are the retained tokens. We show the visualization results of four tasks on four LIBERO datasets in Figure 13. It shows the retained tokens are the task-relevant and dynamic tokens.

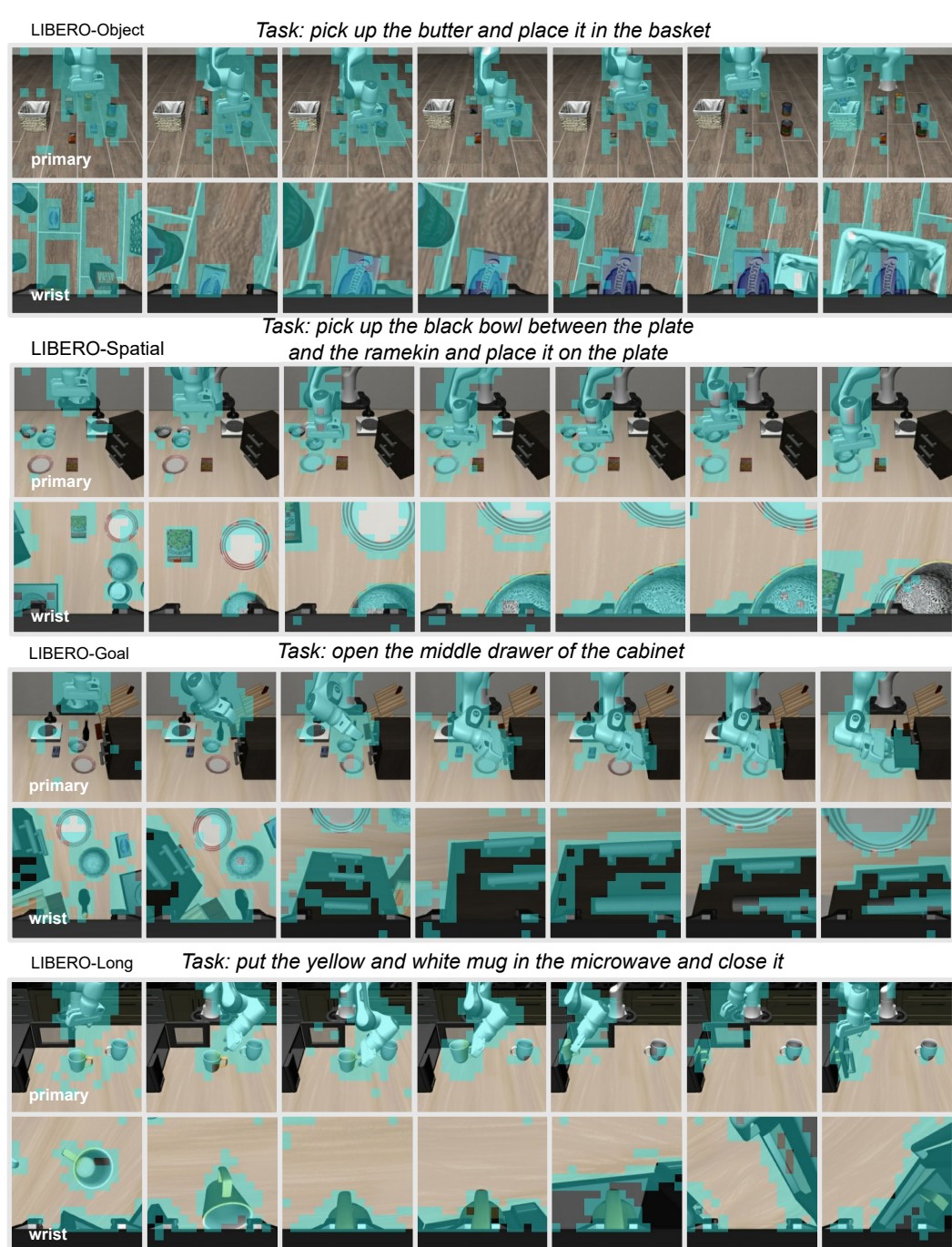

Figure 13: Retained tokens during tasks across four datasets and different views.

### A.6.5 REAL WORLD EXPERIMENT

**Finetuning**  Hyper parameters for OpenVLA-OFT training on real-world tasks are shown in Table 12. We train until the mean L1 loss between predicted and ground-truth normalized actions (scaled between [-1,+1]) falls below 0.01. We decay the learning rate from 5e-4 to 5e-5 after 50K steps.

| Hyperparameter | Value |
|---|---|
| # GPUs | 8 × NVIDIA H200 (141GB VRAM) |
| Learning rate (LR) | 5e-4 (decays to 5e-5 after 50K steps for all tasks) |
| Total batch size | 64 (8 per GPU) |
| # Train steps | "Pick up the cube and place it in the pan.": 80K |
| | "Remove the cube from the tube rack.": 100K |
| | "Pick up the purple cube and put it in the pot.": 80K |
| | "Lift the cup from the tray.": 80K |
| Input images | 1 third-person + 1 side view + 1 wrist camera |
| Image size | 224 × 224 px |
| Obs. history | Single-step (no history) |
| LoRA rank | 32 |
| Action chunk size | 25 steps (open-loop execution) |
| Use proprio | Yes |
| Use FiLM | Yes |
| Trainable params | 853M total: 111M (LoRA) + 269M (action head) + 17M (proprio proj.) + 456M (FiLM proj.) |
| Image augmentations | Random crops (90%), color jitter: |
| | — Brightness: ±0.2 |
| | — Contrast: 0.8–1.2 |
| | — Saturation: 0.8–1.2 |
| | — Hue: ±0.05 |
| | — Crop scale/ratio: [0.9, 0.9] / [1.0, 1.0] |

Table 12: OpenVLA-OFT hyperparameters for real-world experiments. Includes parallel decoding, action chunking, continuous actions with L1 regression, and additional inputs (wrist/side cameras + robot state).

**Evaluation Details**  All evaluations were performed on a single NVIDIA RTX 4090 GPU (24GB VRAM) as the inference machine.

Real-world task details:

1. **"Pick up the cube and place it in the pan."**
   - Task: Use the single-arm robot to grasp the cube and place it in the pan
   - Dataset: 52 demonstrations (52 training)
   - Episode length: 10576 timesteps (353 seconds)
2. **"Remove the cube from the tube rack."**
   - Task: Use the single-arm robot to grasp the tube and carefully place it on the desk
   - Dataset: 50 demonstrations (50 training)
   - Episode length: 10397 timesteps (347 seconds)
3. **"Pick up the purple cube and put it in the pot."**
   - Task: Use the single-arm robot to grasp the purple cube and place it into the pot
   - Dataset: 56 demonstrations (56 training)
   - Episode length: 17129 timesteps (571 seconds)
4. **"Lift the cup from the tray."**
   - Task: Use the single-arm robot to reach for and lift the cup with a stable grasp
   - Dataset: 51 demonstrations (51 training)
   - Episode length: 11788 timesteps (393 seconds)

Evaluation result: See Figure 14

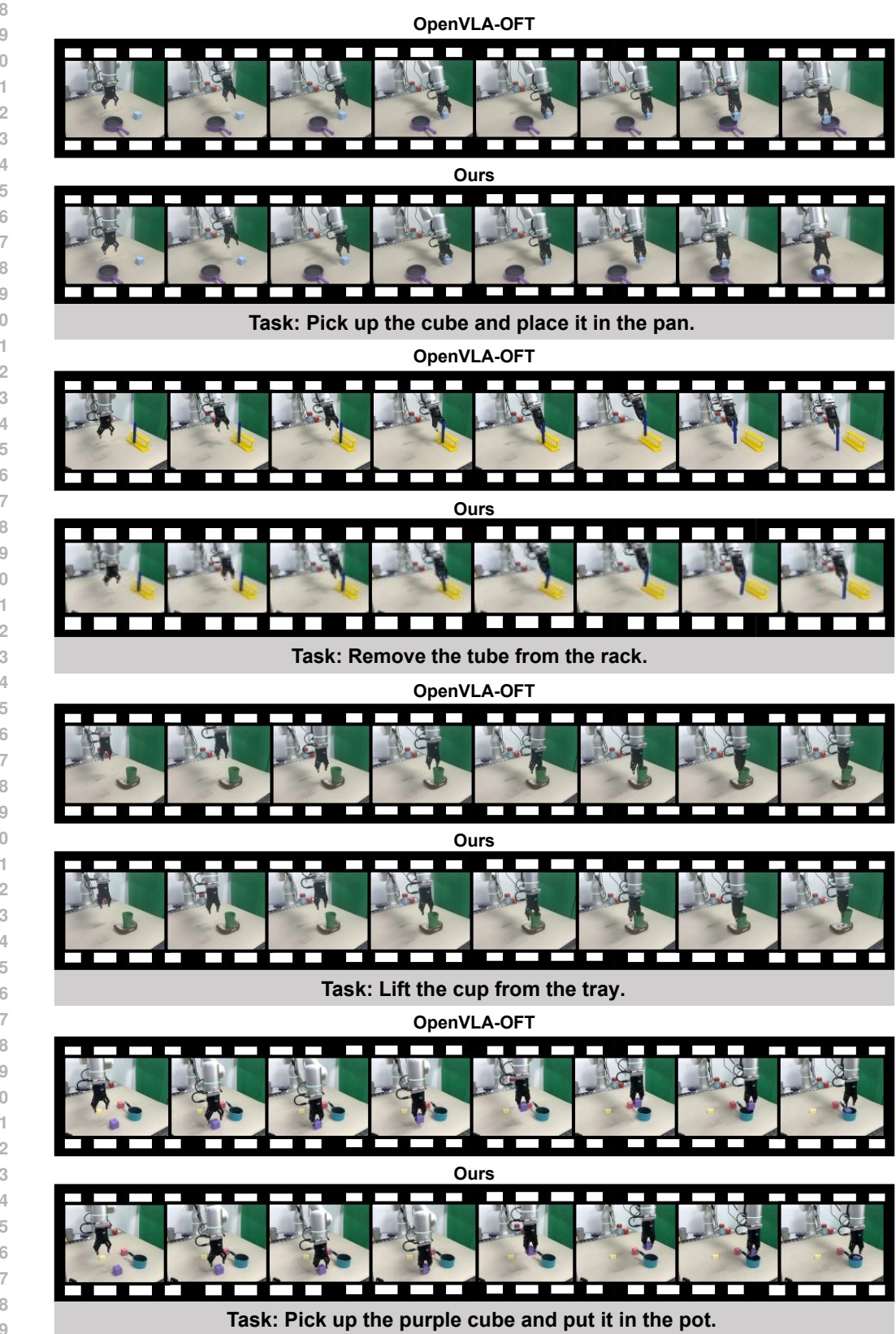

Figure 14: The real-world experiment visualization of OpenVLA-OFT and our method across various tasks.

