# OpenReview forum: "SpecPrune-VLA: Accelerating Vision-Language-Action Models via Action-Aware Self-Speculative Pruning"
_ICLR.cc/2026/Conference — Submitted to ICLR 2026_

### Official Review · Reviewer_fdcZ · 2025-10-14

**Soundness:** 3
**Presentation:** 2
**Contribution:** 2
**Rating:** 2
**Confidence:** 3

**Summary:**

The authors propose SpecVLA - an acceleration method for Vision-Language-Action Models. The acceleration is obtained by A)  Action-level static pruning: the authors compute an attention based score on the visual and text tokens from the middle and the last layer, then the top K visual tokens are chosen, based on the score values. The authors add top-k dynamic tokens, if the corresponding cosine similarity is greater than a threshold value. This combination of tokens is also carried forward to the next step, if two frames are quite similar. B) Layer pruning: The attention scores are modified to include a metric for token confidence. Attention entropy is calculated and is modified to consider a metric for layer confidence. A final importance score for the tokens is calculated to help dynamic pruning. C) Action aware controller: The model shifts tasks based on the rotational and translational velocity.

**Strengths:**

A> The method is training free

B> The method proposes a multi-level pruning strategy, thereby helping the user to choose the computational requirements.

C> The pipeline combines local and global context  and uses that to make the end to end pipeline efficient.

**Weaknesses:**

A> The novelty of the method is limited. The proposed contributions have appeared in other contexts earlier. For instance: Attention entropy in LLMs to ensure faster inference [1], choosing transformer layers and tokens based on context [2], calculating token importance score [3], combining local and global features for inference [4], using translation and rotation velocities to determine tasks [5].

B> In equation 4, the term “rank-based weight” is vague. It is not clear if it is a concatenation of the earlier scores or is a smaller network.

C> There are a lot of threshold and design choices that need clarifications to understand the optimal choice, for instance: the value of top-K tokens, velocity thresholds, dynamic token threshold. The appendix includes instances of tokens pruned - but it does not show a trade-off between performance and latency, to understand the optimal values.

D> The paper has limited experimental results - the pipeline has been evaluated on a single dataset, but multiple devices. Since there is an intersection of scope with a lot of other efficient transformer works, the paper lacks comparison with other contemporary efforts.


[1] Zhang, Zhisong, et al. "Attention Entropy is a Key Factor: An Analysis of Parallel Context Encoding with Full-attention-based Pre-trained Language Models." CoRR (2024).

[2] Meng, Lingchen, et al. "Adavit: Adaptive vision transformers for efficient image recognition." Proceedings of the IEEE/CVF conference on computer vision and pattern recognition. 2022.

[3] Xu, Xuwei, et al. "GTP-ViT: efficient Vision transformers via graph-based token propagation." Proceedings of the IEEE/CVF Winter Conference on Applications of Computer Vision. 2024.

[4] Norouzi, Narges, et al. "Algm: Adaptive local-then-global token merging for efficient semantic segmentation with plain vision transformers." Proceedings of the IEEE/CVF Conference on Computer Vision and Pattern Recognition. 2024.

[5] Qu, Delin, et al. "SpatialVLA: Exploring Spatial Representations for Visual-Language-Action Model." CoRR (2025).

**Questions:**

The paper is an interesting read. I have the following questions:

A>The paper focuses on different varieties of a single dataset: LIBERO. It would be great if the authors could explain the scalability of the approach. I understand from the website of the dataset - that every variety has 10 tasks. Is there a possibility that the accelerations are limited to the lower number of tasks?

B>How do you calculate “rank-based weight” in equation 4?

---

> ### Author Response · Authors · 2025-11-22
> **Official Response to Reviewer fdcZ**
>
> We thank you for your reviews and will address your concerns as follows:
>
> **Weakness A: " The novelty of the method is limited. The proposed contributions have appeared in other contexts earlier. For instance: Attention entropy in LLMs to ensure faster inference [1], choosing transformer layers and tokens based on context [2], calculating token importance score [3], combining local and global features for inference [4], using translation and rotation velocities to determine tasks [5]."**
>
> Thank you for providing these references. We have carefully viewed the cited works and would like to acknowledge that our method indeed draws upon some existing concepts, such as attention entropy and attention scores. Our dynamic pruning method is motivated by the idea in paper[1] that "High entropy indicates a dispersed and unstable attention distribution, causing the model to fail in effectively retrieving information from the context." **However, our primary technical approach and motivation are distinct from prior work [2],[3],[4],[5].**
>
>
> ### **Our Distinct Motivation**
>
> First of all, our motivation is the special characteristics of VLA tasks: we observed the Temporal Attention Redundancy. Consistant inference share highly similar environmental features(e.g. background, object layouts). When the input is highly similar(text input is completely the same), the attention maps of the whole model should remain largely the same. We also conduct experiment and found that in different tasks and during the whole task process, the overlap of top-k important tokens between consecutive inferences remain in a high level(65%~95%). We define the Recall of Previous Important Tokens, which measures the overlap between the globally important token set at the previous step $V_{t-1}$ and the current set $V_t$, normalized by the size of the current set.
> Formally, it is expressed as:
> $$
>     \text{Recall}(V_{t-1}, V_t) = \frac{|V_{t-1} \cap V_t|}{|V_t|}
> $$
> We observe that this recall reaches an average of **73\%** throughout the task execution, indicating strong temporal persistence in token relevance. Therefore, we can reuse the global information.
>
>
> **vs. [4] (Algm - Local/Global Features):**
> Our "global" feature refers to the information of the whole model and "local" refers to the partial model (a few layers). This is fundamentally different from spatial "local/global" features within a single image, as used in Algm.
>
> **vs. [2] (AdaViT - Layer-wise Pruning):**
> AdaViT requires **additional training** of a decision network to select tokens or skip layers. In contrast, our Layer-level Dynamic Pruning is **training-free**. It leverages intrinsic layer confidence (measured by attention entropy) to adaptively refine token importance as features propagate through the model, making it suitable for generative VLA tasks without extra overhead.
>
> **vs. [3] (GTP-ViT - Importance Score):**
> GTP-ViT's importance score measures a token's ability to propagate information in a graph structure. Moreover, it is calculated within layers. Our score directly measures task relevance by combining image-to-text cross-attention and layer-specific confidence. This enables us to identify and preserve tokens that are globally important across the model depth, ensuring critical semantic and task-relevant features are retained for robust action generation in VLA models..
>
> **vs. [5] (SpatialVLA - Velocity):**
> SpatialVLA uses action statistics to create adaptive grids for **action tokenization**. Our method uses speed signals as a controller: preserving more visual tokens for high-precision (fine-grained) actions and pruning more during coarse movements, ensuring efficiency without sacrificing critical accuracy. This action-aware dynamic pruning is key to **efficient control**.
>
> In summary， we presents a cohesive and efficient solution specifically tailored for the challenges of VLA real-time robotic inference, offering a distinct approach compared to existing works in both methodology and motivation.

---

> ### Author Response · Authors · 2025-11-22
> **Official Response to Reviewer fdcZ(2)**
>
> **QA>: "The paper focuses on different varieties of a single dataset: LIBERO. It would be great if the authors could explain the scalability of the approach. I understand from the website of the dataset - that every variety has 10 tasks. Is there a possibility that the accelerations are limited to the lower number of tasks?"**
>
> Theoratically, the acceleration rate has a lower bound for all tasks: The most token number after pruning for every image is: $N_{pruned}$ = $K_{dynamic}$ + $K_{global}$ + 2*$K_{local}$ = 88. In total, the worst case reduce input tokens by about 51.6%. This guaranteed a minimum speedup of 1.38×. Therefore, the accelerations is inherently generalizable across different task suites.
>
> To validate the scalability of our approach, we also conduct experiments on LIBERO-plus[1] benchmark, which includes far more and challenging tasks (10000 tasks in total). Due to the time and resource limit, we randomly choose **200** different tasks that adding perturbations to camera viewpoint, light condition, background and noise.
> |Method| Camera | Light | Background | Noise| Average SR (%)|
> |:---:|:---:|:---:|:---:|:---:|:---:|
> |OpenVLA-OFT| 96.0% | 88.5% | 95.0% | 97.0% | 94.1%
> |Ours (prune rate=0.8)| 94.0% | 87.8% | 95.0% | 97.0% | 93.5%
>
> **Table 1**: Performance comparison on LIBERO-plus tasks.
>
> Our method maintains strong performance in various cases. This confirms that SpecPrune-VLA generalizes well to various tasks types and environmental perturbations.
>
> **Weakness&Question B> In equation 4, the term “rank-based weight” is vague. It is not clear if it is a concatenation of the earlier scores or is a smaller network.**
>
> The rank-based weight is not a smaller network. It is defined as eq.5  $\omega^{\text{rank}, l}_i = \frac{\sigma(\text{rank}^{l}_i)}{\sum_j \sigma(\text{rank}^{l}_j)}$ ,calculated directly from the token's attention ranking within a layer $l$. This allows the model give a high score to most critical tokens without introducing additional learnable parameters or complex architectures.
>
> [1]Fei, Senyu, et al. "LIBERO-Plus: In-depth Robustness Analysis of Vision-Language-Action Models." arXiv preprint arXiv:2510.13626 (2025).

---

> ### Author Response · Authors · 2025-11-22
> **Official Response to Reviewer fdcZ(3)**
>
> **Weakness C> There are a lot of threshold and design choices that need clarifications to understand the optimal choice, for instance: the value of top-K tokens, velocity thresholds, dynamic token threshold. The appendix includes instances of tokens pruned - but it does not show a trade-off between performance and latency, to understand the optimal values.**
>
> Thank you for raising this important point regarding the design choices and their optimization. We acknowledge that our initial submission did not sufficiently clarify the rationale behind our key hyperparameters. Below, we provide a more focused analysis on the trade-offs between performance (Success Rate, SR) and efficiency (Speedup), which guided our optimal selections.
>
> ### **The selection top-Ks' value**
> Our base K values are chosen to maximize the temporal consistency of critical information, measured by the Recall of Previous Important Tokens. $\text{Recall}(V_{t-1}, V_t) = \frac{|V_{t-1} \cap V_t|}{|V_t|}.$
>
> * For the $K_{global}$, we evaluated value of 10, 20, 30 and observed the average recall rate is the highest when $K_{global}$ equals 20.
> * For $K_{local}$ and $K_{dynamic}$, we tested several pairs: (12, 15), (24, 20), (36, 30). The token set $\hat V_t$ supplemented by local and dynamic tokens, reach the highest recall rate $\text{Recall}(V_{t-1}, \hat V_t) = \frac{|V_{t-1} \cap \hat V_t|}{|\hat V_t|}$ when he pair equals (24, 20). Therefore, we set the base value $K_{global}$=20, $K_{local}$=24 and $K_{dynamic}$=20.
>
> The detail result are as follows.
>
> |||||||||
> |:---:|:---:|:---:|:---:|:---:| :---: |:---:|:---:|
> | $K_{global}$| 10 | **20** |30 | $(K_{local}$, $K_{dynamic})$ | (12, 10) | **(24, 20)** | (36, 30)|
> |  $\text{Recall}(V_{t-1}, V_t) $| 56% | **73%** | 68% | $\text{Recall}(V_{t-1},  \hat V_t)$ | 68% | **92%** | 87% |
>
> **Table2** The recall rate of different $K$ values.
>
> This high recall indicates that these values effectively capture the persistent, task-relevant visual features across consecutive steps, forming a robust foundation for static pruning. The choice prioritizes preserving model accuracy by ensuring globally important context is retained.
>
> ### **Optimal Prune Rate α: Balancing Accuracy and Speed**
> The prune rate is a parameter multiplied to basic K value ($K_i = \alpha \cdot K_i$, where $K_i$ is the base value mentioned above). It is the primary lever for trading off speed and accuracy. As demonstrated in the ablation study, decreasing α increases speedup but degrades success rate. Our choice of α=0.8 strikes an optimal balance:
>
> | Prune rate α | LIBERO-Spatial | LIBERO-Object | LIBERO-Goal | LIBERO-10 | Average SR (%) | Average Speedup |
> |:---:|:---:|:---:|:---:|:---:|:---:|:---:|
> |None(OpenVLA-OFT)| 97.6 / 1.00 $\times$ | 96.5 / 1.00 $\times$ | 97.9 / 1.00 $\times$ |94.5 / 1.00 $\times$| 96.6 | 1.00 $\times$ |
> | 1.0 | 97.6 / 1.41 $\times$ | 96.3 / 1.43 $\times$ | 97.9 / 1.40 $\times$ | 94.0 / 1.38 $\times$ | 96.5 | 1.40 $\times$ |
> | **0.8** | **97.6 / 1.44 $\times$** | **95.8 / 1.46 $\times$** | **97.7 / 1.48 $\times$** | **93.4 / 1.44 $\times$** | **96.1** | **1.45 $\times$** |
> | 0.6 | 97.4 / 1.52 $\times$ | 93.2 / 1.50 $\times$ | 96.0 / 1.52 $\times$ | 90.2 / 1.51 $\times$ | 94.2 | 1.51 $\times$ |
>
> **Table3** :The performance (Success Rate (%) / Speedup ($\times$)) of different prune rate.
>
> ### **The threshold τ for Dynamic Token Selection**
> The similarity threshold τ filters patches with low change magnitude to avoid false positives caused by lighting or camera noise. Since we select the top-$K_{Dynamic}$ most dissimilar patches among those below τ , the exact value is not highly sensitive. In simulation, we set τ=0.95 ; in real-world settings with higher noise, we use τ=0.8 . Experiments show performance remains stable within a range of τ∈[0.9,0.99] in simulation and τ∈[0.6,0.9] in real world, confirming robustness.
>
> In summary, our design choices are data-driven: base K values are optimized for accuracy and temporal consistency, α is tuned for the best speed-accuracy trade-off, and τ is selected for robustness.

---

> ### Author Response · Authors · 2025-11-22
> **Official Response to Reviewer fdcZ(4)**
>
> **Weakness D> The paper has limited experimental results - the pipeline has been evaluated on a single dataset, but multiple devices. Since there is an intersection of scope with a lot of other efficient transformer works, the paper lacks comparison with other contemporary efforts.**
>
> In simulation environment, we evaluated on LIBERO, which is a popular benchmark. We also evaluated our methods on real world-task (Sec.7.6) and achieve 1.70 $\times$ speedup without obvious performance drop.
>
> To further validate our method's generalization and robustness, we have conducted experiments on LIBERO-plus that adding perturbations to ***camera viewpoint,light condition, background and noise***.
>
> |Method| Camera | Light | Background | Noise| Average SR (%)|
> |:---:|:---:|:---:|:---:|:---:|:---:|
> |OpenVLA-OFT| 96.0% | 88.5% | 95.0% | 97.0% | 94.1%
> |Ours (prune rate=0.8)| 94.0% | 87.8% | 95.0% | 97.0% | 93.5%
>
> **Table4**:  Performance comparison on LIBERO-plus tasks. (Same as Table1)
>
> The result reveals that our method has strong roboustness towards perturbations. Our global information reuse is effective in different camera viewpoints, which also reveals that temporal-consistency is a distinct characteristic of VLA task. Our velocity-based sample and local information help in camera-noise and light-changing situition.
>
> For the second point, there are some works in VLM and ViT, we have chosen SparseVLM as a comparision. We also introduce results of other two advanced pruning method in VLM: FastV(ECCV24)[1] and DivPrune(CVPR25)[2]. Our method in prune rate = 0.8 setting retain about 60% visual tokens on average. Therefore, for fair comparision, we retain 60% tokens for the two methods.
>
> | Method | LIBERO-Spatial | LIBERO-Object | LIBERO-Goal | LIBERO-10 | Average SR (%) | Average Speedup |
> |:---:|:---:|:---:|:---:|:---:|:---:|:---:|
> |OpenVLA-OFT| 97.6% | 96.5%  | 97.9%  |94.5% | 96.6% | 1.00 $\times$ |
> | FastV | 94.6%  | 95.8% | 94.0% | 88.8% | 93.3% | 1.44 $\times$ |
> | DivPrune | 92.4%  | 91.2%  | 89.0%  | 84.8% | 89.4% | 1.46 $\times$ |
> | **Ours(prune rate=0.8)** | **97.6%** | **95.8%** | **97.7%** | **93.4%** | **96.1%** | **1.45 $\times$** |
>
> **Table5:** Comparision with contemporary works on ViT.
>
> The results shows that our method outperforms FastV and DivPrune in success rate while achieving similar speedup.
> * FastV prunes tokens based on early-layer prefill attention, being training-free but ignoring global context.
> * DivPrune selects diverse tokens to reduce redundancy, but its feature-space diversity objective overlooks the dynamic, task-critical relevance of tokens in VLA models, making it less suitable for fine-grained spatial reasoning.
>
> [1]Chen, Liang, et al. "An image is worth 1/2 tokens after layer 2: Plug-and-play inference acceleration for large vision-language models." European Conference on Computer Vision. Cham: Springer Nature Switzerland, 2024.
>
> [2]Alvar, Saeed Ranjbar, et al. "Divprune: Diversity-based visual token pruning for large multimodal models." Proceedings of the Computer Vision and Pattern Recognition Conference. 2025.
>
> We agree that further analysis can strengthen the work. We will incorporate the detailed data from the LIBERO-plus experiments ，the ablation studies on hyperparameter sensitivity and comparision with contemporary works into the revised manuscript to provide a more comprehensive presentation of our method's robustness and optimal configuration.
>
> We hope that our detailed responses meet with your approval.

---

> ### Comment · Reviewer_fdcZ · 2025-11-24
> **Response to comments by authors**
>
> To ensure proper credit, the authors should explicitly cite paper [1] when discussing the use of attention entropy. Additionally, the manuscript would benefit from a clearer explanation of how the proposed method's specific features distinguish it from [1].
>
> If there is limited room for these additional results, please add these ablation studies in the appendix.
>
> Thank you for providing these additional insights from your work! I do not have any further questions.

---

> ### Author Response · Authors · 2025-11-25
>
> We are grateful for your positive re-evaluation and thank you for raising score to 6. We will explicitly cite paper[1] when discussing attention entropy and add the above ablation study in the revised version.

---

### Official Review · Reviewer_zpp5 · 2025-10-31

**Soundness:** 3
**Presentation:** 3
**Contribution:** 3
**Rating:** 6
**Confidence:** 3

**Summary:**

This paper addresses the inference latency bottleneck in Vision-Language-Action (VLA) models, primarily due to the LLM backbone. The paper presents a well-motivated, comprehensive pruning strategy tailored to VLA models. The primary strengths are its training-free design and the novel action-aware controller, though the complexity and tuning of the multi-stage pruning process are minor weaknesses.

**Strengths:**

- The paper is built on clear insights specific to the robotics domain: VLA inference is compute-bound, visual information is temporally redundant across action steps, and not all actions are equally sensitive to pruning.

- The lightweight action-aware controller is practical. It uses end-effector velocity to heuristically differentiate between coarse- and fine-grained actions, reducing pruning during high-precision phases. The ablation study confirms that this controller is critical for recovering the success rate lost due to aggressive pruning.

- The method is evaluated thoroughly in both simulation on the LIBERO benchmark and on a physical robot. Demonstrating consistent speedups (1.46x-1.70x) with minimal success rate loss across multiple hardware platforms (NVIDIA A800 and RTX 3090)  strongly supports the claims of effectiveness and generalization.

**Weaknesses:**

- Concern about the reproducibility and generalizability of the pruning configuration. The method relies on several hyperparameters ($K_G$, $K_D$, $K_L$, $\beta$, $L_{prune}$, etc.), notably the overall pruning ratio $\alpha$, which is set manually for each of the four LIBERO task suites. It is unclear how $\alpha$ should be determined for a new, unseen task without iterative tuning.

- The complexity of the two-level pruning interaction might be high. Tokens are first pruned statically at the action level ($V_{retain} = V_{global} \cup V_{dynamic} \cup V_{local}$), and this retained set is then pruned dynamically at the layer level. This multi-stage filtering, combined with the action-aware controller, makes the final token count difficult to track and analyze.

- The velocity-based frame sampling strategy appears heuristic, with constants chosen empirically. While effective, the paper lacks analysis of how sensitive performance is to these hyperparameters or whether this strategy generalizes across robot morphologies or task distributions.

**Questions:**

- How sensitive is the performance to the choice of temporal window T in the dynamic frame comparison? Would an adaptive method (e.g., using visual change detection) yield better results? Also, the method reuses attention scores from layers 15 and 32 of the prior step. Why these specific layers? Is there a systematic way to select “global” layers, or is this also empirical?

- The ablation in Table 2 implies the controller adds 1.5ms (72.3ms vs 70.8ms). What is the total latency cost of calculating $V_{local}$ (requiring partial forward passes), $V_{dynamic}$ (patch similarity), and the layer-wise entropy scores?

- EfficientVLA is noted to be originally designed for **diffusion-based action heads**, yet it’s compared against OpenVLA-OFT (which uses an MLP head). While the authors acknowledge this, the performance gap (e.g., 24.4% SR drop in LIBERO-Long) may partly stem from architectural mismatch rather than pruning quality alone.

---

> ### Author Response · Authors · 2025-11-22
> **Official Response to Reviewer zpp5**
>
> **Q1>: "How sensitive is the performance to the choice of temporal window T in the dynamic frame comparison? Would an adaptive method (e.g., using visual change detection) yield better results? Also, the method reuses attention scores from layers 15 and 32 of the prior step. Why these specific layers? Is there a systematic way to select “global” layers, or is this also empirical?"**
>
> We appreciate this insightful question and we will address your concerns as follows:·
>
> ### **1. Choice of temporal window T**
>
> The performance is indeed sensitive to the choice of temporal window $T$. We observed that a fixed T can be suboptimal under varying motion dynamics. A small $T$ (e.g., 1) captures fast motion but may be noisy(Figure4(b)), while a large $T$ smooths out short-term changes but risks missing rapid dynamics. If we set T to a fixed value such as 1, we observed success rate sometimes drop by >15% because of missing real dynamic tokens. Therefore we propose velcity-based frame sampling and comparision, it is an adaptive method.
>
> Our key insight is that the optimal T depends on motion dynamics and the eef's speed is a good representative of changing intensity. Since end-effector speed v linearly correlates with scene change intensity, we set  $T = \left\lfloor b + k \cdot v \right\rfloor + 4$ , with k=−1, b=7. Parameters are empirically chosen by observing optimal T at extreme speed values.
>
> As shown in Figure 9, A.5.4, our method consistently focus on dynamic elements during the whole task procee. Our experiment on real robot that achieves 1.70 speedup without obvious drop in accuracy also shows the robustness of velocity-based frame sample.
>
> ### **2. Regarding visual change detection**
> Using visual change detection to dynamically select the reference frame can potentially yield better results. For instance, we can maintain a "keyframe" and only update it when the embedding difference exceeds a threshold:
> Update if $\frac{1}{N} \sum_i \| z_t^i - z_{\text{key}}^i \| > \tau$. This is a reliable method to extract the most appropriate frame to compare with. However, the method needs calculating similarity every timestep to detect visual change, increasing latency and computation.
>
> In fact, our selection of window T is an adaptive method, it relys on the eef's speed and the speed can well indicates the changing intensity.
>
> ### **3. The selection of reuse layers**
>
> Our layer selection is motivated by prior findings in both language and vision-language models, which show that different transformer layers capture distinct types of semantic and spatial information. In LLMs, early layers typically encode low-level linguistic features, middle layers handle compositional semantics, and deeper layers contribute to task-specific reasoning (e.g., Eagle[1]). In VLMs, works such as [2] reveals that early layers preserve rich local spatial details, while deeper layers extend to global semantic relationships.
> In VLA models, we observe a hierarchical representation pattern across layers.
>
> Through empirical analysis on VLA model, we find that:
>
> Layers 1–2 primarily attend to general image semantics and object categories.
> Shallow layers (3–10) focus more on static background elements and global scene layout.
> Middle (around 15) and deep (around 30–32) layers progressively refine attention toward task-critical regions—such as target objects and interaction points—while maintaining contextual coherence.
> Given this, we choose to extract global importance from a combination of middle and deep layers (specifically the 15th and 32nd) to balance early-stage semantic awareness with late-stage task specificity. This design ensures robust identification of persistently important tokens across consecutive inference steps.

---

> ### Author Response · Authors · 2025-11-22
> **Official Response to Reviewer zpp5(2)**
>
> ### **Experiment on reuse layer selection**
> To rigorously evaluate the impact of layer choice on pruning effectiveness, we conducted a post-hoc ablation study that separates the analysis of token importance from real-time execution. Specifically, we first allow the full model to perform a complete forward pass without executing any actions in the environment. Then using these precomputed attention scores, we apply top-K pruning based on attention weights from different layer configurations (e.g., shallow, middle, deep), retain only the selected tokens, and re-run the inference with the pruned input to generate new actions. By comparing the resulting success rates under various layer selections and pruning ratios, we can systematically assess which layers provide the most reliable importance signals for token retention.
>
> |||Selected layers to reuse|||
> | :---: | :---: | :---: |:---: |:---: |
> |prune token number|shallow layers (e.g.6)| shallow&middle (e.g.6&31) | shallow&deep (e.g.6&31) | **middle&deep  (e.g.15&31)** |
> |None|96.2%|96.2%|96.2%|**96.2%**|
> |25%|94.8%|94.8%|96.2%|**96.2%**|
> |50%|92.0%|94.0%|94.0%|**96.2%**|
> |75%|88.0%|92.0%|92.0%|**95.6%**|
>
> **Table1**: Performance of different reuse layer selection under different prune token number.
>
> As shown in Table 6, using shallow layers (e.g., layer 6) or a combination of shallow and middle layers for token reuse leads to noticeable performance degradation as the pruning ratio increases, especially beyond 50%. In contrast, reusing information from middle and deep layers (e.g., layers 15 & 31) maintains high success rates even under aggressive pruning (75%), demonstrating that deeper representations provide more reliable and semantically meaningful importance scores. This makes the middle&deep configuration the optimal choice for our method.
>
> [1]Li, Yuhui, et al. "Eagle: Speculative sampling requires rethinking feature uncertainty." arXiv preprint arXiv:2401.15077 (2024). \
> [2]Liu, Yiming, Yuhui Zhang, and Serena Yeung-Levy. "Mechanistic Interpretability Meets Vision Language Models: Insights and Limitations." The Fourth Blogpost Track at ICLR 2025.
>
>
> **Q2> The ablation in Table 2 implies the controller adds 1.5ms (72.3ms vs 70.8ms). What is the total latency cost of calculating (requiring partial forward passes),  (patch similarity), and the layer-wise entropy scores?**
>
> Here are the breakdown on every module.
> | $V_{Dynamic}$ Patch similarity|$V_{Local}$ (first 2 layers)| Attention Entropy $H$ |  Our method|Original Model |
> | :--: |:--: |:--: |:--: |:--: |
> |1.8 $\pm$ 0.1ms|5.5 $\pm$ 0.2ms| 1.0 $\pm$ 0.2ms| 70~75ms| 109.0ms |
>
> **Complexity Analysis**
> * Patch Similarity: The cosine similarity between corresponding patches in two frames is computed as
> $\text{Sim}(P_m^{i,j}, P_n^{i,j}) = \frac{P_m^{i,j} \cdot P_n^{i,j}}{\| P_m^{i,j}\|_2 \| P_n^{i,j}\|_2}$ , For N patches and patch size p ,this operation has complexity $\mathcal{O}(N \cdot p^2)$. Here N=256, p=14, which are not a big value.This operation adds negligible overhead(2.4%) to the model。
> * As defined in Eq. (6), The attention entropy is computed over the $N×N$ attention matrix at layer $l$. Its complexity is $\mathcal{O}(N^2)$ and introduces 1ms latency. Naively computing it at every step would incur ~32ms overhead. To avoid this, we observe that entropy patterns are stable across inference steps within the same task. Therefore, we compute the layer confidence scores only once during the first inference and reuse them in subsequent steps.
> * The complexity of single transformer layer is $\mathcal{O}(LD^2+L^2D)$, where $L$ is the sequence length, $D$ is the hidden dimension. The partial forward of $V_{Local}$ isn't an additional spending in calculation, it is a necessary process in the model inference with unpruned input tokens. We leverage it's calculation result of attention.

---

> ### Author Response · Authors · 2025-11-22
> **Official Response to Reviewer zpp5 (3)**
>
> **Weakness1>: Concern about the reproducibility and generalizability of the pruning configuration. The method relies on several hyperparameters ($K_{Dynamic}$ , $K_{Global}$ , $K_{Local}$ , $L_{Prune}$, $\beta$ etc.), notably the overall pruning ratio $\alpha$ is set manually for each of the four LIBERO task suites. It is unclear how should be determined for a new, unseen task without iterative tuning.**
>
> We appreciate the reviewer’s insightful question regarding hyperparameter configuration. We would like to emphasize that parameters' selection is guided by ablation studies and quantitative evaluation, rather than arbitrary tuning.
>
> ### **Base K-values are selected based on recall performance**
> From Figure 3II, we observe that up to 192 visual tokens can be pruned with negligible success rate degradation, indicating that retaining approximately 60–100 task-relevant tokens is sufficient for effective inference. Under this insight, we conducted ablation studies to select the optimal base values (**Table3**):
>
> * For the $K_{global}$, we evaluated value of 10, 20, 30 and observed the average recall rate is the highest when $K_{global}$ equals 20.
> * For $K_{local}$ and $K_{dynamic}$, we tested several pairs: (12, 15), (24, 20), (36, 30). The token set $\hat V_t$ supplemented by local and dynamic tokens, reach the highest recall rate $\text{Recall}(V_{t-1}, \hat V_t) = \frac{|V_{t-1} \cap \hat V_t|}{|\hat V_t|}$ when he pair equals (24, 20). Therefore, we set the base value $K_{global}$=20, $K_{local}$=24 and $K_{dynamic}$=20.
>
> |||||||||
> |:---:|:---:|:---:|:---:|:---:| :---: |:---:|:---:|
> | $K_{global}$| 10 | **20** |30 | $(K_{local}$, $K_{dynamic})$ | (12, 10) | **(24, 20)** | (36, 30)|
> |  $\text{Recall}(V_{t-1}, V_t) $| 56% | **73%** | 68% | $\text{Recall}(V_{t-1}, V^*_t)$ | 68% | **92%** | 87% |
>
> **Table2**: The recall rate of different $K$
>
> ### **Ablation study on prune ratio $\alpha$**
> The prune rate $\alpha$ serves as a high-level knob to balance speed and accuracy. It controls the global K -values $K_{global}$ ,$K_{local}$ and $K_{dynamic}$ by scaling: $K_i= \alpha \cdot K_i$. To evaluate the robustness of performance under different hyperparameter configurations, we conducted an ablation study across the LIBERO benchmark with varying α . Results are summarized below:
>
> | Prune rate α | LIBERO-Spatial | LIBERO-Object | LIBERO-Goal | LIBERO-10 | Average SR (%) | Average Speedup |
> |:---:|:---:|:---:|:---:|:---:|:---:|:---:|
> |None(OpenVLA-OFT)| 97.6 / 1.00 $\times$ | 96.5 / 1.00 $\times$ | 97.9 / 1.00 $\times$ |94.5 / 1.00 $\times$| 96.6 | 1.00 $\times$ |
> | 1.0 | 97.6 / 1.41 $\times$ | 96.3 / 1.43 $\times$ | 97.9 / 1.40 $\times$ | 94.0 / 1.38 $\times$ | 96.5 | 1.40 $\times$ |
> | **0.8** | **97.6 / 1.44 $\times$** | **95.8 / 1.46 $\times$** | **97.7 / 1.48 $\times$** | **93.4 / 1.44 $\times$** | **96.1** | **1.45 $\times$** |
> | 0.6 | 97.4 / 1.52 $\times$ | 93.2 / 1.50 $\times$ | 96.0 / 1.52 $\times$ | 92.2 / 1.51 $\times$ | 94.7 | 1.51 $\times$ |
>
> **Table4:** Ablation study on prune rate $\alpha$.
>
> Our approach exhibits low sensitivity to α : even under aggressive pruning (α=0.6 ), the average success rate remains above 94%, with minimal degradation compared to the unpruned baseline. This indicates strong robustness across hyperparameter configurations. To achieve an **optimal trade-off** between speed and accuracy, we set α=0.8 , which yields an average speedup of 1.45$\times$ while maintaining a success rate of 96.1%.

---

> ### Author Response · Authors · 2025-11-22
> **Official Response to Reviewer zpp5(4)**
>
> **The threshold τ for Dynamic Token Selection** \
> The similarity threshold τ filters patches with low change magnitude to avoid false positives caused by lighting or camera noise. Since we select the $top-K_{Dynamic}$ most dissimilar patches among those below τ , the exact value is not highly sensitive. In simulation, we set τ=0.95 ; in real-world settings with higher noise, we use τ=0.8 . Experiments shows performance remains stable within a range of τ∈[0.9,0.99] in simulation and τ∈[0.6,0.9] in real world, confirming robustness.
>
> **Decay Rate $\beta$ and target pruning layer $L_{prune}$ in Dynamic pruning**
> * Emprically, in exponential moving average (EMA) formulations of the form $S_i^{(l)} = (1 - \beta) \cdot S_i^{(l-1)} + \beta \cdot s_i^{(l)}$ , update rate is often set to a small value to ensure smooth calculation. In our method, $\beta$ can be set in a loose range (0.1~0.3) without performance loss. However, large value(>0.6) will cause loss in task success rate。  This is because overly aggressive updates amplify noise from early layers, causing unimportant tokens to be incorrectly assigned high importance scores, while critical tokens may be mistakenly pruned.
> * Regarding the choice of target pruning layer $L_{prune}$,  we select non-consecutive layers at varying depths. This design ensures that token importance scores are sufficiently refined through multiple layers of contextual updates before pruning is applied. Pruning at consecutive layers, in contrast, risks premature removal of tokens before their representations mature, leading to loss of crucial semantic information in deeper layers.
> To validate this, we conducted ablation experiments by applying layer-level pruning on two consecutive block ranges: layers (5–10) and (25–30). In both cases, the model’s task success rate dropped by more than 20%, confirming that consecutive pruning severely disrupts the flow of visual reasoning and harms final performance.
>
> ### **Generalization ability to unseen tasks**
>
> We also do experiment on LIBERO-plus[1] to validate the generalization. LIBERO-plus includes far more and challenging tasks (10000 tasks in total). Due to the time and resource limit, we randomly choose **200** different tasks that adding perturbations to ***camera viewpoint, light condition, background and noise***. Using a fixed α=0.8 and above parameters, the results are:
> |Method| Camera | Light | Background | Noise| Average Success Rate (%)|
> |:---:|:---:|:---:|:---:|:---:|:---:|
> |OpenVLA-OFT| 96.0% | 88.5% | 95.0% | 97.0% | 94.1%
> |Ours (prune rate=0.8)| 94.0% | 87.8% | 95.0% | 97.0% | 93.5%
>
> **Table4:** Evaluation on LIBERO-plus.
>
> These results confirm that our configuration generalizes well to unseen conditions without iterative optimization.
>
> **Q3>EfficientVLA is noted to be originally designed for diffusion-based action heads, yet it’s compared against OpenVLA-OFT (which uses an MLP head). While the authors acknowledge this, the performance gap (e.g., 24.4% SR drop in LIBERO-Long) may partly stem from architectural mismatch rather than pruning quality alone.**
>
> EfficientVLA additionally prunes a fixed number of layers, which may cause structural degradation in VLA models. In LLMs, layer pruning is a common model compression technique because language generation is discrete and more tolerant to errors. However, due to the continuous nature of action generation, layer pruning has limited generalization in VLA scenarios.
>
> For a more comprehensive comparison, we integrate the state-of-the-art VLM pruning methods FastV(ECCV24)[1] and DivPrune(CVPR25)[2] for evaluation.
> | Method | LIBERO-Spatial | LIBERO-Object | LIBERO-Goal | LIBERO-10 | Average SR (%) | Average Speedup |
> |:---:|:---:|:---:|:---:|:---:|:---:|:---:|
> |OpenVLA-OFT| 97.6% | 96.5%  | 97.9%  |94.5% | 96.6% | 1.00 $\times$ |
> | FastV | 94.6%  | 95.8% | 94.0% | 88.8% | 93.3% | 1.44 $\times$ |
> | DivPrune | 92.4%  | 91.2%  | 89.0%  | 84.8% | 89.4% | 1.46 $\times$ |
> | **Ours(prune rate=0.8)** | **97.6%** | **95.85** | **97.7%** | **93.4%** | **96.1%** | **1.45 $\times$** |
>
> **Table5:** Comparision with contemporary works on ViT.
>
> The results shows that our method outperforms FastV and DivPrune in success rate while achieving similar speedup.
> * FastV prunes tokens based on early-layer prefill attention, being training-free but ignoring global context.
> * DivPrune selects diverse tokens to reduce redundancy, but its feature-space diversity objective overlooks the dynamic, task-critical relevance of tokens in VLA models, making it less suitable for fine-grained spatial reasoning.
>
> [1]Alvar, Saeed Ranjbar, et al. "Divprune: Diversity-based visual token pruning for large multimodal models." Proceedings of the Computer Vision and Pattern Recognition Conference. 2025. \
> [2]Chen, Liang, et al. "An image is worth 1/2 tokens after layer 2: Plug-and-play inference acceleration for large vision-language models." European Conference on Computer Vision. Cham: Springer Nature Switzerland, 2024.

---

### Official Review · Reviewer_UQji · 2025-10-31

**Soundness:** 3
**Presentation:** 3
**Contribution:** 3
**Rating:** 6
**Confidence:** 2

**Summary:**

The paper proposes a pruning method for VLA models. The authors diagnose and show that current VLA acceleration is compute-bound, so it’s intuitive to prune tokens in LLM, thus redundant visual patches as nearby images in the query are often similar in background and importance. They also incorporate global information from the previous step to guide pruning and introduce an action-aware controller adapter. Experiments show that their approach achieves higher speedup than prior methods while achieving better task success rates.

**Strengths:**

1. The diagnosis is intuitive, especially the demonstration that now accelerations are bound by compute. Thus pruning image tokens is really intuitive, especially since there are redundancies in images.
2. The results are convincing as higher speedup and better success rate are shown in the experiment section.
3. The paper is well written, and the explanation of the method is clear.

**Weaknesses:**

1. The method mostly relies on hand-tuned heuristics (velocity thresholds, entropy weighting, pruning ratios), it would be great to see extended ablation studies.
2. Do experiments in VLA pruning usually not report success rates and confidence intervals? This can be hard to judge statistical significance.

**Questions:**

Please see the weakness section.

---

> ### Author Response · Authors · 2025-11-24
> **Official response to Reviewer UQji**
>
> **Weakness 1:"The method mostly relies on hand-tuned heuristics (velocity thresholds, entropy weighting, pruning ratios), it would be great to see extended ablation studies."**
>
> We appreciate the reviewer’s insightful question regarding hyperparameter configuration. While certain parameters in our framework—such as the top-K values and the global prune rate α are determined empirically, we emphasize that their selection is guided by ablation studies and quantitative evaluation, rather than arbitrary tuning.
>
> ### **Base K-values are selected based on recall performance**
>
> First, we define the ***Recall of Previous Important Tokens***, which measures the proportion of currently important tokens that were also identified as important in the previous step.
> Formally, given the globally important token set at time t−1 , denoted $V_{t-1}$ and the current set $V_t$ , we compute:
> $$
>     \text{Recall}(V_{t-1}, V_t) = \frac{|V_{t-1} \cap V_t|}{|V_t|}
> $$
>
> From Figure 3II, we observe that up to 192 visual tokens can be pruned with negligible success rate degradation, indicating that retaining approximately 60–100 task-relevant tokens is sufficient for effective inference. Under this insight, we conducted ablation studies to select the optimal base values (**Table1**):
>
> * For the $K_{global}$, we evaluated value of 10, 20, 30 and observed the average recall rate is the highest when $K_{global}$ equals 20.
> * For $K_{local}$ and $K_{dynamic}$, we tested several pairs: (12, 15), (24, 20), (36, 30). The token set $\hat V_t$ supplemented by local and dynamic tokens, reach the highest recall rate $\text{Recall}(V_{t-1}, \hat V_t) = \frac{|V_{t-1} \cap \hat V_t|}{|\hat V_t|}$ when he pair equals (24, 20). Therefore, we set the base value $K_{global}$=20, $K_{local}$=24 and $K_{dynamic}$=20.
>
> |||||||||
> |:---:|:---:|:---:|:---:|:---:| :---: |:---:|:---:|
> | $K_{global}$| 10 | **20** |30 | $(K_{local}$, $K_{dynamic})$ | (12, 10) | **(24, 20)** | (36, 30)|
> |  $\text{Recall}(V_{t-1}, V_t) $| 56% | **73%** | 68% | $\text{Recall}(V_{t-1}, V^*_t)$ | 68% | **92%** | 87% |
>
> **Table1**: The recall rate of different $K$
>
> ### **Ablation on prune ratio $\alpha$**
> The prune ratio $\alpha$ serves as a high-level knob to balance speed and accuracy. It controls the global K -values $K_{global}$ ,$K_{local}$ and $K_{dynamic}$ by scaling: $K_i= \alpha \cdot K_i$. To evaluate the robustness of performance under different hyperparameter configurations, we conducted an ablation study across the LIBERO benchmark with varying α . Results are summarized below:
>
> | Prune ratio α | LIBERO-Spatial | LIBERO-Object | LIBERO-Goal | LIBERO-10 | Average SR (%) | Average Speedup |
> |:---:|:---:|:---:|:---:|:---:|:---:|:---:|
> |None(OpenVLA-OFT)| 97.6 / 1.00 $\times$ | 96.5 / 1.00 $\times$ | 97.9 / 1.00 $\times$ |94.5 / 1.00 $\times$| 96.6 | 1.00 $\times$ |
> | 1.0 | 97.6 / 1.41 $\times$ | 96.3 / 1.43 $\times$ | 97.9 / 1.40 $\times$ | 94.0 / 1.38 $\times$ | 96.5 | 1.40 $\times$ |
> | **0.8** | **97.6 / 1.44 $\times$** | **95.8 / 1.46 $\times$** | **97.7 / 1.48 $\times$** | **93.4 / 1.44 $\times$** | **96.1** | **1.45 $\times$** |
> | 0.6 | 97.4 / 1.52 $\times$ | 93.2 / 1.50 $\times$ | 96.0 / 1.52 $\times$ | 92.2 / 1.51 $\times$ | 94.7 | 1.51 $\times$ |
>
> **Table2:** Ablation study on prune ratio $\alpha$.
>
> Even under aggressive pruning (α=0.6 ), the average success rate remains above 94%, with minimal degradation compared to the unpruned baseline. This indicates strong robustness across hyperparameter configurations. To achieve an optimal trade-off between speed and accuracy, we set α=0.8 , which yields an average speedup of 1.45$\times$ while maintaining a success rate of 96.1%.

---

> ### Author Response · Authors · 2025-11-24
> **Official response to Reviewer UQji(2)**
>
> ### **The threshold τ for Dynamic Token Selection**
> The similarity threshold τ filters patches with low change magnitude to avoid false positives caused by lighting or camera noise. Since we select the $top-K_{Dynamic}$ most dissimilar patches among those below τ , the exact value is not highly sensitive. In simulation, we set τ=0.95 ; in real-world settings with higher noise, we use τ=0.8 . Experiments shows performance remains stable within a range of τ∈[0.9,0.99] in simulation and τ∈[0.6,0.9] in real world, confirming robustness.
>
> ### **Entropy weighting clarification**
> We would like to clarify that the entropy weight $ω_{conf} = 1/(H̄ + ϵ)$ is not a hand-tuned heuristic, but a dynamic signal computed from the model's internal attention entropy $H̄$ at each inference step. This layer-specific confidence score is automatically derived from the model’s own state, making it an adaptive, data-driven mechanism rather than a manually set parameter. The Update Rate $\beta$ is a empirical value and we do the ablation in the following section.
>
> #### **Ablation on entropy-based layer weight**
>
> We conduct extended ablation study on entropy-based layer weight as follows, the recall rate represents the overlap between final retain token and ground-truth important token set.
>
> ||Recall rate(%)|\|\|||| Success rate(%)  |||
> |:-:|:-:|:-:|:-:|:-:| :-:|:-:|:-:|
> |Method| $\text{Recall}(V_{t-1}, V_t)$ | \|\| |LIBERO-Spatial | LIBERO-object | LIBERO-goal|  LIBERO-10 |Average SR |
> |Entropy-based | 88% | \|\| | 97.6% | 96.3% | 97.7% | 94.0% | 96.4% |
> |Average weighting | 66% | \|\| |92.0% | 92.4% | 95.0% | 88.6% | 92.0% |
>
> From the perspective of recall rate, the average weighting mistakenly prune globally important tokens, thus achievingb poorer success rate, especiallt in challenging long-horizon tsks. It utilize uncertain information from high-entropy layer, prune important tokens and cause cumulative error.
>
> **Update Rate $\beta$ exploration**
> * Emprically, in exponential moving average (EMA) formulations of the form $S_i^{(l)} = (1 - \beta) \cdot S_i^{(l-1)} + \beta \cdot s_i^{(l)}$ , update rate is often set to a small value to ensure smooth calculation. In our method, $\beta$ can be set in a loose range (0.1~0.3) without performance loss. However, large value(>0.6) will cause loss in task success rate。  This is because overly aggressive updates amplify noise from early layers, causing unimportant tokens to be incorrectly assigned high importance scores, while critical tokens may be mistakenly pruned.
>
> ### **Ablation on velocity threshold**
> As stated in Section 6.2 of the paper, our core insight is that the operation process of a VLA can be categorized into fine-grained or coarse-grained modes based on its speed. In the fine-grained mode, the z-axis velocity is negative, and both translational and rotational velocities are small. We use predefined thresholds to determine if a mode transition has occurred.
>
> Here, we need to clarify that our velocity and velocity thresholds are set based on the normalized delta actions(speed) of the VLA model's output. This output is independent of the specific robot arm's configuration or dynamic characteristics. The normalization standard is derived from the min-max values used in the model's training dataset, which is a necessary step for training VLA models and thus not an additional requirement imposed by our method. This normalization ensures the generality and broad applicability of our approach.
>
> Expired by the empirical data, we set the final threshold as $v_t^{th}$=0.5, $v_r^{th}$=0.2. if $v_t < v^{th}_
> t, v_r < v^{th}_r$ and $∆z≤0$. We also provide the ablation study on the threshold on LIBERO-spatial.
>
> | $v_t^{th}$ |  0.3 | 0.5 | 0.7 | $v_r^{th}$ | 0.1 | 0.2 | 0.4 |
> | :--: | -- | -- | --| :--: | -- | -- | --|
> | Success rate| 96.8% | 97.4% | 97.6%| \||96.8% | 97.4% | 97.4%|
>
> The result shows that when the threshold value is too small, the majority part of the action will be classified as "coarse-grained," leading to aggressive pruning, which may cause a noticeable drop in performance (e.g., success rate drops to 95.4%). Conversely, when the threshold value is too large, nearly the entire process is classified as "fine-grained," resulting in more tokens being retained. The marginal gain in success rate is minimal and it introduces additional latency. Therefore, our chosen thresholds ($v_t^{th}$=0.5, $v_r^{th}$=0.2) achieve an optimal balance, ensuring high task success while maximizing acceleration.
>
> Moreover, we also explored the performance of our action controller in extreme cases. We set prune rate to 0.4, which means only 15%-20% image is retained. We tested in LIBERO-long.
> | w / w\o action adapter | With action adapter | Without action adapter|
> | :--: | :--: | :--: |
> | Success rate| 74.0% | 84.4% |
>
> The results demonstrate that the action adapter maintains high performance even under extreme conditions. It helps the model to see less but act smarter.

---

> ### Author Response · Authors · 2025-11-24
> **Official response to Reviewer UQji(3)**
>
> ### **Generalization of action adapter to various environment and robot**
> We also observe the task process of UniVLA, it is employed on WidowX robot arm in SimplerEnv simulation environment. The actions of every steps are as follows.
> | Step | Trans. Speed (norm) | Z-speed (norm) | Rot. Speed (norm) | Action Mode |
> |------|:--------------------:|:----------------:|:-------------------:|:-------------------:
> | 1    | 0.86               | -0.77          | 0.99              |
> | 2    | 0.85               | -0.87          | 0.92              |
> | 3    | 0.78               | -0.92          | 0.78              |
> | 4    | 0.76               | -0.94          | 0.66              |
> | 5    | 0.69               | -0.88          | 0.54              |
> | 6    | 0.62               | -0.80          | 0.43              |
> | 7    | 0.58               | -0.75          | 0.36              |
> | 8    | 0.54               | -0.70          | 0.28              |
> | 9    | 0.52               | -0.68          | 0.24              |
> | **10**   | **0.48**               |**-0.63**          | **0.20**               | **fine-grained**      |
> | **11**   | **0.45**               | **-0.59**          | **0.16**              | **fine-grained**      |
> | **12**   | **0.38**               | **-0.50**          | **0.12**              | **fine-grained**      |
> | **13**   | **0.30**               | **-0.40**          | **0.07**              | **fine-grained**      |
> | **14**   | **0.05**               | **-0.03**          | **0.01**              | **fine-grained**      |
> | **15**   | **0.39**               | **0.49**          | **0.18**              | **fine-grained**     |
> | 16   | 0.67               | 0.83           | 0.33              |
> | 17   | 0.79               | 0.91           | 0.47              |
> | 18   | 0.89               | 0.94           | 0.47              |
> | 19   | 0.95               | 0.89           | 0.39              |
> | 20   | 0.95               | 0.67           | 0.25              |
> | 21   | 0.90               | 0.37           | 0.13              |
> | 22   | 0.84               | 0.11           | 0.07              |
>
>
> The data reveals a universal pattern: a phase of high-speed movement (Step 1-9) corresponds to coarse-grained actions like approaching the target object. This transitions sharply into a sustained period of low translational and rotational speeds, accompanied by a negative z-axis displacement (Step 10-14), which perfectly aligns with the fine-grained manipulation phase (e.g., grasping). Finally, the action concludes with another high-speed phase (Step 15-22) for transferring and placing.
>
> Due to the limitation of Openreview, we can't perform it in diagram. However, we'll add the diagram into the revised version. The data shows that our method can correctly recogonize coarse/fine grained action in different models and environment.
>
> ****
>
> We also do experiment on LIBERO-plus[1] with the above parameter and prune rate=0.8 to validate the generalization. LIBERO-plus includes far more and challenging tasks (10000 tasks in total). Due to the time and resource limit, we randomly choose **200** different long-horizon and object-recognition tasks that adding perturbations to ***camera viewpoint, light condition, background and noise***. Using a fixed α=0.8 , the results are:
> |Method| Camera | Light | Background | Noise| Average Success Rate (%)|
> |:---:|:---:|:---:|:---:|:---:|:---:|
> |OpenVLA-OFT| 96.0% | 88.5% | 95.0% | 97.0% | 94.1%
> |Ours (prune rate=0.8)| 94.0% | 87.8% | 95.0% | 97.0% | 93.5%
>
> **Table3:** Evaluation on LIBERO-plus. (Same with Table2)
>
> Our method shows strong robustness and generalization ability in dynamic and tougher environment.
>
> [1]Fei, Senyu, et al. "LIBERO-Plus: In-depth Robustness Analysis of Vision-Language-Action Models." arXiv preprint arXiv:2510.13626 (2025).
>
> **Weakness 2:"Do experiments in VLA pruning usually not report success rates and confidence intervals? This can be hard to judge statistical significance."**
>
> In our experiments, each task is repeated 50 times (same as OpenVLA-OFT) to reduce random variation. We use fixed environment seeds in simulation, which is a common practice for reproducibility. It makes results deterministic and enables fair comparison.
>
> While many VLA studies do not report confidence intervals, we agree they strengthen evaluation. To address this, we are currently running additional experiments with multiple random seeds to compute meaningful confidence intervals. Due to the high computational cost, this process takes time, but we will share the results as soon as they are ready.

---

> > ### Comment · Reviewer_UQji · 2025-11-26
> >
> > I appreciate the author's response. My concerns have been solved. I am happy to maintain the original rating and continue to recommend accepting the paper.

---

### Official Review · Reviewer_rhEd · 2025-11-01

**Soundness:** 2
**Presentation:** 2
**Contribution:** 3
**Rating:** 4
**Confidence:** 3

**Summary:**

The paper presents SpecPrune-VLA, a training-free acceleration scheme for Vision-Language-Action (VLA) models that prunes visual tokens using action-level static pruning that reuses global attention statistics from the previous action step and supplements them with local early-layer attention and motion-based dynamic tokens. They also adopt payer-level dynamic pruning driven by rank-weighted attention and per-layer attention-entropy and a lightweight action-aware controller that reduces pruning during fine-grained manipulation and increases it during coarse motion.
The evaluation on LIBERO tasks shows 1.46$\times$ average end-to-end speedup over OpenVLA-OFT and 1.70$\times$ on real robot.

**Strengths:**

- The efficiency for VLA is essential, as it highly impacts the performance of real robot controlling systems.
- The proposed two-level pruning, combining global and local information, is well-motivated and beneficial.
- The coarse/fine-grained switching based on real scenarios is helpful to improve the performance.

**Weaknesses:**

- The paper lacks an analysis of the importance/confidence scores. How are the layer confidences distributed?
- Ablation in Tab.2 is incomplete. e.g., there is no ablation on global only vs local only, entropy-based layer confidence vs rank-only weighting.
- Some annotations are potentially confusing. For example,  line-227 $I_1,I_2$ might be clearer if written as $I_t,I_{t+1}$

**Questions:**

- Sec 3
  - How many tokens per image in total in Fig.3?
- Sec 4
  - In Eq.1, how are the layers selected?
  - In Eq.1, what does the subscript  $i$ in $V_i$ denote?
  - In $T=\lfloor b + k\cdot v \rfloor + 4$, both translational velocity and rotational velocity seem to be included in $v$. Are these two types of velocity handled in the same way, or is there a difference in how they are applied?
  - For sec.4.1.2, could you discuss the computational complexity and the overhead introduced by the method?
  - Have you tried different similarity metrics?

---

> ### Author Response · Authors · 2025-11-24
> **Official response to Reviewer rhEd**
>
> Thanks for your review and we will address your concern as follow
>
> **Q1: "Sec 3: How many tokens per image in total in Fig.3?"**
>
> There are 256 tokens per image. The vision encoder convert every image into 256 token embeddings and send into the model backnone. Generally, VLA models encode images into 256 tokens. Some researches has explored to encode images into more patches such as 324, but there isn't an obviouse rise in performance.
>
> **Q2&3: "Sec 4: In Eq.1, how are the layers selected? In Eq.1, what does the subscript i in $V_i$ denote?"**
>
> In Eq.1:
> $\text{Score}_l(V_i)=$
>
> $\frac{1}{H \cdot m} \sum_{h=1}^{H} \sum_{j=1}^{m} A_l^h(V_i, t_j)$
>
> We do not select specific layers; rather, Eq.1 is computed for any single layer $l$ independently.
>
> The subscript $i$ in $V_i$ denotes the index of a specific visual token within the sequence of visual tokens extracted from the input image.
>
> To clarify further:
> - After processing the image through the vision encoder (e.g., DINOv2), the image features are flattened into a sequence of visual tokens: $V_1, V_2, ..., V_N$, where $N$ is the total number of tokens.
> - Each $V_i$ represents one such token, typically corresponding to a patch of the original image.
> - The score $\text{Score}_l(V_i)$ measures how strongly token $V_i$ attends to the task instruction text tokens in layer $l$. A higher score indicates that the model considers $V_i$ more relevant to the current task at that layer.
>
> Further, I guess you are asking how are the global-reuse layers selected. The following are our explanation:
>
> **The selection of reuse layers**
>
> Our layer selection is motivated by prior findings in both language and vision-language models, which show that different transformer layers capture distinct types of semantic and spatial information. In LLMs, early layers typically encode low-level linguistic features, middle layers handle compositional semantics, and deeper layers contribute to task-specific reasoning (e.g., Eagle[1]). In VLMs, works such as [2] reveals that early layers preserve rich local spatial details, while deeper layers extend to global semantic relationships.
> In VLA models, we observe a hierarchical representation pattern across layers.
>
> Through empirical analysis on VLA model, we find that:
>
> Layers 1–2 primarily attend to general image semantics and object categories.
> Shallow layers (3–10) focus more on static background elements and global scene layout.
> Middle (around15) and deep (around 30) layers progressively refine attention toward task-critical regions such as target objects and interaction points.
> Given this, we choose to extract global importance from a combination of middle and deep layers (specifically the 15th and 32nd) to balance early-stage semantic awareness with late-stage task specificity. This design ensures robust identification of persistently important tokens across consecutive inference steps.
>
> ### **Experiment on reuse layer selection**
> To rigorously evaluate the impact of layer choice on pruning effectiveness, we conducted a post-hoc ablation study that separates the analysis of token importance from real-time execution. Specifically, we first allow the full model to perform a complete forward pass without executing any actions in the environment. Then using these precomputed attention scores, we apply top-K pruning based on attention weights from different layer configurations (e.g., shallow, middle, deep), retain only the selected tokens, and re-run the inference with the pruned input to generate new actions. By comparing the resulting success rates under various layer selections and pruning ratios, we can systematically assess which layers provide the most reliable importance signals for token retention.
>
> |||Selected layers to reuse|||
> | :---: | :---: | :---: |:---: |:---: |
> |prune token number|shallow layers (e.g.6)| shallow&middle (e.g.6&31) | shallow&deep (e.g.6&31) | **middle&deep  (e.g.15&31)** |
> |None|96.2%|96.2%|96.2%|**96.2%**|
> |25%|94.8%|94.8%|96.2%|**96.2%**|
> |50%|92.0%|94.0%|94.0%|**96.2%**|
> |75%|88.0%|92.0%|92.0%|**95.6%**|
>
> **Table1**: Performance of different reuse layer selection under different prune token number.
>
> As shown in Table 6, using shallow layers or a combination of shallow and middle layers for token reuse leads to noticeable performance degradation as the pruning ratio increases, especially beyond 50%. In contrast, reusing information from middle and deep layers maintains high success rates even under aggressive pruning (75%), demonstrating that deeper representations provide more reliable and semantically meaningful importance scores. This makes the middle&deep configuration the optimal choice for our method.
>
> [1]Li, Yuhui, et al. "Eagle: Speculative sampling requires rethinking feature uncertainty." arXiv preprint arXiv:2401.15077 (2024).
> [2]Liu, Yiming, Yuhui Zhang, and Serena Yeung-Levy. "Mechanistic Interpretability Meets Vision Language Models: Insights and Limitations." The Fourth Blogpost Track at ICLR 2025.

---

> ### Author Response · Authors · 2025-11-24
> **Official response to Reviewer rhEd(2)**
>
> **Q4: "In $T = \left\lfloor b + k \cdot v \right\rfloor + 4$, both translational velocity and rotational velocity seem to be included in. Are these two types of velocity handled in the same way, or is there a difference in how they are applied?"**
>
> Thank you for the insightful question. We would like to clarify that in the equation $T=⌊b+k⋅v⌋+4$ , the velocity v refers only to the translational velocity of the end-effector, and does not include rotational velocity.
>
> While both translational and rotational motions contribute to overall movement, we find that rotational velocities are typically small and involve subtle changes in pose, which do not induce significant pixel-level variations in the input images. As a result, they are less effective as indicators of visual change intensity. We'll clarify this in our revised version.
>
> **Q5: "For sec.4.1.2, could you discuss the computational complexity and the overhead introduced by the method?"**
>
> Sure. \
> **Complexity Analysis**
> * Patch Similarity:  The cosine similarity is calculated based on the raw patches before the image being encoded. The similarity between corresponding patches in two frames is computed as
> $\text{Sim}(P_m^{i,j}, P_n^{i,j}) = \frac{P_m^{i,j} \cdot P_n^{i,j}}{\| P_m^{i,j}\|_2 \| P_n^{i,j}\|_2}$ , For N patches and patch size p ,this operation has complexity $\mathcal{O}(N \cdot p^2)$. Here N=256, p=14, which are not a big value.
>
> * Transformer layer: Each transformer layer process L tokens, with hidden size D. The complexity is $ \mathbf{O}(LD^2+L^2D)$. In our 7B model, D=4096.
>
> Here is the breakdown on the module.
> | Patch similarity| Whole Model (our method) |
> | :--: |:--: |
> |1.8 $\pm$ 0.1ms|70~75ms|
>
> It shows that calculating patch similarity adds negligible overhead (2.4%) to the overall model.
>
> **Q6: "Have you tried different similarity metrics?"**
>
> We also tried L1/L2 distances. While computationally efficient, they are highly sensitive to absolute intensity variations (e.g., lighting changes or sensor noise), leading to large numerical differences even when the underlying content remains similar. The threshold is also hard to set. This limits their generalization across diverse scenes and conditions.
>
> Cosine similarity naturally maps the angle between vectors to the range [−1,1] , which is typically non-negative and bounded in [0,1] when features are normalized. It measures directional consistency rather than magnitude, making it invariant to changes in illumination or contrast. This normalization effect enhances stability and improves generalization across diverse scenes.

---

> ### Author Response · Authors · 2025-11-25
>
> **Weakness 1: "The paper lacks an analysis of the importance/confidence scores. How are the layer confidences distributed?"**
>
> We thank the reviewer for this insightful comment. We agree that a deeper analysis of the confidence scores is essential to justify our design choices.
>
> Inspired by information theory and prior work on attention interpretability [1], we interpret low entropy as indicative of high certainty in token selection, where the model focuses on a small set of dominant tokens. In contrast, high entropy suggests diffuse attention and uncertain semantic encoding, implying that the layer may contribute less reliable signals for downstream decisions.
>
> Here, we normalize the attention entropy by the maximum possible entropy($log(N_{VisualTokens})$). Empirically,The normalized entropy across layers ranges approximately from 0.5 to 0.99, 0.4 indicates highly concentrated attention and 0.9 indicates nearly uniform attention distribution, with 0.7 serving as a meaningful threshold:
> ||||Attention| Entropy||||
> |:---:|:---:|:---:|:---:|:---:|:---:|:---:|:---:|
> | **Layer 0** | **Layer 1** | **Layer 2** | **Layer 3** | **Layer 4** | **Layer 5** | **Layer 6** | **Layer 7** |
> | 0.9991 | 0.9369 | 0.8848 | 0.9668 | 0.9515 | 0.9269 | 0.8783 | 0.8899 |
> | **Layer 8** | **Layer 9** | **Layer 10** | **Layer 11** | **Layer 12** | **Layer 13** | **Layer 14** | **Layer 15** |
> | 0.8451 | 0.8474 | 0.8377 | 0.7928 | 0.7964 | 0.7775 | 0.7759 | 0.7596 |
> | **Layer 16** | **Layer 17** | **Layer 18** | **Layer 19** | **Layer 20** | **Layer 21** | **Layer 22** | **Layer 23** |
> | 0.6569 | 0.7008 | 0.7586 | 0.6195 | 0.5279 | 0.5277 | 0.5891 | 0.6480 |
> | **Layer 24** | **Layer 25** | **Layer 26** | **Layer 27** | **Layer 28** | **Layer 29** | **Layer 30** | **Layer 31** |
> | 0.4788 | 0.5741 | 0.6248 | 0.6140 | 0.7716 | 0.5637 | 0.7914 | 0.8143 |
>
> **Table:** Attention entropy across layers
>
> As shown in the table, there is a clear trend: early layers exhibit higher entropy, indicating they attend more diffusely — often capturing background or context information, while later layers show lower entropy, reflecting increasingly focused and task-relevant attention patterns. Based on this observation, we design the layer confidence score as an inverse function of entropy, assigning higher weights to more certain (low-entropy) layers.
>
> We include a detailed ablation study in the next section to demonstrate the impact of this design choice on overall performance. We will also include a heatmap visualization to clearly illustrate the layer-wise entropy distribution
> in the revised version of the paper.
>
> [1] Zhang, Zhisong, et al. "Attention Entropy is a Key Factor: An Analysis of Parallel Context Encoding with Full-attention-based Pre-trained Language Models." CoRR (2024).

---

> ### Author Response · Authors · 2025-11-25
> **Official response to Reviewer rhEd(4)**
>
> **Weakness 2: Ablation in Tab.2 is incomplete. e.g., there is no ablation on global only vs local only, entropy-based layer confidence vs rank-only weighting.**
>
> Thank you for this valuable feedback. We agree that our initial ablation study in Table 2 was incomplete.
>
> ### **Ablation on Global attention reuse**
>
> Firstly, to quantify the overlap of globally important tokens between consecutive inference, we define the ***Recall of Previous Important Tokens***, which measures the proportion of currently important tokens that were also identified as important in the previous step.
> Formally, given the globally important token set at time t−1 , denoted $V_{t-1}$ and the current set $V_t$ , we compute:
> $$
>     \text{Recall}(V_{t-1}, V_t) = \frac{|V_{t-1} \cap V_t|}{|V_t|}
> $$
>
> We conducted an ablation study comparing Recall rate and success rate of pruning with and without global attention information. For fair comparision, when pruning with global information, $K_{local}$ is set to 44, otherwise, we use $K_{local}$=24, $K_{global}$=20. Therefore, the speedup of two cases are the same.
>
> ||Recall rate(%)|\|\|||| Success rate(%)  |||
> |:---:|:---:|:---:|:---:|:---:| :---:|:---:|:---:|
> |w/ and w/o| $\text{Recall}(V_{t-1}, V_t)$ | \|\| |LIBERO-Spatial | LIBERO-object | LIBERO-goal|  LIBERO-10 |Average SR |
> |w/ global attn. | 92% | \|\| | 97.6% | 96.3% | 97.7% | 94.0% | 96.4% |
> |w/o global attn. | 84% | \|\| |95.6% | 94.8% | 95.0% | 90.0% | 93.4% |
>
> **Table1**: Ablation study on global reuse on LIBERO.
>
> As shown, incorporating global context yields  better token recall and higher success rates, demonstrating that prior-step attention maps provide valuable, task-consistent guidance that purely local methods cannot fully capture. This supports our key insight: due to the spatio-temporal coherence in robotic tasks, the global attention map from the previous step serves as a reliable prior for the current step.
>
> ### **Ablation on entropy-based layer weight**
>
> In Sec.5, our layer confidence score is defined as $\omega_{\text{conf}}^{(l)} = \frac{1}{\bar{H}^{(l)} + \epsilon}$, where $\bar{H}^{(l)}$ is the attention entropy of layer $l$ and $\epsilon > 0$ is a small constant for numerical stability. The low entropy indicates high certainty in token selection. The high entropy layer indicates that the information of the layer is uncertain therefore we should lay less confidence on the layer.
>
> We conduct ablation study on entropy-based layer weight as follows, the recall rate represents the overlap between final retain token and ground-truth important token set.
>
> ||Recall rate(%)|\|\|||| Success rate(%)  |||
> |:---:|:---:|:---:|:---:|:---:| :---:|:---:|:---:|
> |Method| $\text{Recall}(V_{t-1}, V_t)$ | \|\| |LIBERO-Spatial | LIBERO-object | LIBERO-goal|  LIBERO-10 |Average SR |
> |Entropy-based | 88% | \|\| | 97.6% | 96.3% | 97.7% | 94.0% | 96.4% |
> |Average weighting | 66% | \|\| |92.0% | 92.4% | 95.0% | 88.6% | 92.0% |
>
> From the perspective of recall rate, the average weighting mistakenly prune globally important tokens, thus achievingb poorer success rate, especiallt in challenging long-horizon tsks. It utilize uncertain information from high-entropy layer, prune important tokens and cause cumulative error.
>
> **Weakness3: Some annotations are potentially confusing. For example, line-227 $I_1, I_2$
>  might be clearer if written as $I_t, I_{t+1}$**
>
> We have reviewed the entire paper and will revise the confusing annotations.
>
> We hope that our responses have adequately addressed your concerns. Thank you again for the insightful feedback, which has helped us improve the clarity and rigor of our work.

---

### Official Review · Reviewer_eraN · 2025-11-01

**Soundness:** 2
**Presentation:** 2
**Contribution:** 2
**Rating:** 4
**Confidence:** 4

**Summary:**

The paper introduces SpecPrune-VLA, a training-free two-level token pruning framework designed to accelerate inference in VLA models. It leverages global context reuse across consecutive action steps and local speculative attention to select important visual tokens, while a lightweight action-aware controller dynamically adjusts pruning aggressiveness based on robot end-effector velocity. Experiments on the LIBERO benchmark and real-world robotic platforms show that SpecPrune-VLA achieves up to 1.57× speedup (A800 GPU) and 1.7× real-world speedup with negligible task success rate degradation (<1%) compared to OpenVLA-OFT.

**Strengths:**

1. Improving the efficiency of VLA is an important research topic.
2. The writing of the paper is quite clear and easy to follow.
3. The method is training-free, making it easily integrable into existing VLA pipelines.

**Weaknesses:**

1. The mathematical formulation of global/local fusion and the entropy-confidence weighting is heuristic.

2. Too many hyperparameters are involved in the proposed method. The pruning heavily depends on multiple manually tuned thresholds (α, τ, β, K values).

3. Experiments are limited to OpenVLA-OFT.

**Questions:**

1. How do you ensure that reusing global attention maps from previous steps does not propagate outdated or erroneous importance information, especially in tasks with dynamic camera motion or sudden object appearance?
2. Many parameters (τ, α, β, K-values, pruning thresholds) are hand-tuned. Have you tested the robustness of performance under different hyperparameter configurations? How sensitive is your approach to these values?
3. The velocity-based controller seems tailored to a specific robot arm setup. How general is this to other robotic systems or action spaces?
4. For 4.1.1, the attention from the 15th and 32nd layer are selected to indicate the performance. What is the rationale? Have you tried other layers?

---

> ### Author Response · Authors · 2025-11-22
> **The Official response to Reviewer eraN**
>
> Thank you for your review and we address your concern as follows:
>
> **Q1: "How do you ensure that reusing global attention maps from previous steps does not propagate outdated or erroneous importance information, especially in tasks with dynamic camera motion or sudden object appearance?"**
>
> We sincerely thank you for this insightful question. We acknowledge that the paper could provide stronger statistical support for our strategy of reusing global attention maps, and we appreciate the opportunity to clarify this important point.
>
> First, the high overlap of important tokens between consecutive inference ensure the errorness is limited. To quantify this overlap, we define the ***Recall of Previous Important Tokens***, which measures the proportion of currently important tokens that were also identified as important in the previous step.
> Formally, given the globally important token set at time t−1 , denoted $V_{t-1}$ and the current set $V_t$ , we compute:
> $$
>     \text{Recall}(V_{t-1}, V_t) = \frac{|V_{t-1} \cap V_t|}{|V_t|}
> $$
> We observe an average recall of 73% across task execution, indicating strong temporal persistence in token relevance, supporting the reliability of leveraging prior-step context.
>
> Moreover, our retained token set is not solely dependent on historical information. It is augmented with two dynamic components: (1) locally important tokens from the current inference (based on early-layer attention), and (2) dynamically changing tokens detected via frame similarity comparison. This fusion ensures that up-to-date visual content is preserved, effectively mitigating the risk of propagating outdated or incorrect importance signals. Importantly, the global attention map is refreshed at every inference step, preventing long-term error accumulation.
>
> ### **Ablation on Global Attention reuse**
> To further validate the benefit of global context reuse, we conducted an ablation study comparing pruning with and without global attention information. For fair comparision, when pruning with global information, $K_{local}$ is set to 44, otherwise, we use $K_{local}$=24, $K_{global}$=20.
>
> |w/ and w/o| $\text{Recall}(V_{t-1}, V^*_t)$|  Avg. Success Rate|
> |:---:|:---:| :---: |
> |With global attn. | 92% |  97.6% |
> |Without global attn. | 84% | 95.3% |
>
> **Table1**: Ablation study on global reuse on LIBERO-spatial and LIBERO-goal.
>
> As shown, incorporating global context yields  better token recall and higher success rates, demonstrating that prior-step attention maps provide valuable, task-consistent guidance that purely local methods cannot fully capture. This supports our key insight: due to the spatio-temporal coherence in robotic tasks, the global attention map from the previous step serves as a reliable prior for the current step.
>
> ### **Dynamic scenarios evaluation**
>
> In scenarios involving camera motion or sudden object appearance, we also conduct experiment on such situitions in LIBERO-plus[1]. We randomly choose **200** tasks hat adding perturbations to **camera viewpoint, light condition, background and noise**.
>
> |Method| Camera | Light | Background | Noise| Average SR (%)|
> |:---:|:---:|:---:|:---:|:---:|:---:|
> |OpenVLA-OFT| 96.0% | 88.5% | 95.0% | 97.0% | 94.1%
> |Ours (prune rate=0.8)| 94.0% | 87.8% | 95.0% | 97.0% | 93.5%
>
> **Table2**: Robustness experiment (Success Rate(SR%)) on LIBERO-plus
>
> The results indicates that the dynamic token and local attention supplementation mechanism ensures critical new information is preserved. Together, these design choices strike an effective balance between stability and adaptability.
>
> [1]Fei, Senyu, et al. "LIBERO-Plus: In-depth Robustness Analysis of Vision-Language-Action Models." arXiv preprint arXiv:2510.13626 (2025).

---

> ### Author Response · Authors · 2025-11-22
> **The Official response to Reviewer eraN(2)**
>
> **Q2: "Many parameters (τ, α, β, K-values, pruning thresholds) are hand-tuned. Have you tested the robustness of performance under different hyperparameter configurations? How sensitive is your approach to these values?"**
>
> We appreciate the reviewer’s insightful question regarding hyperparameter sensitivity. While certain parameters in our framework—such as the top-K values and the global prune rate α are determined empirically, we emphasize that their selection is guided by ablation studies and quantitative evaluation, rather than arbitrary tuning.
>
> ### **Base K-values are selected based on recall performance**
> From Figure 3II, we observe that up to 192 visual tokens can be pruned with negligible success rate degradation, indicating that retaining approximately 60–100 task-relevant tokens is sufficient for effective inference. Under this insight, we conducted ablation studies to select the optimal base values (**Table3**):
>
> * For the $K_{global}$ , we evaluated value of 10, 20, 30 and observed the average recall rate is the highest when $K_{global}$ equals 20.
> * For $K_{local}$ and $K_{dynamic}$ , we tested several pairs: (12, 15), (24, 20), (36, 30). The token set $\hat V_t$  supplemented by local and dynamic tokens, reach the highest recall rate $\text{Recall}(V_{t-1}, \hat V_t) = \frac{|V_{t-1} \cap \hat V_t|}{|\hat V_t|}$ when the pair equals (24, 20). Therefore, we set the base value $K_{global}$=20, $K_{local}$=24 and $K_{dynamic}$=20.
>
> |||||||||
> |:---:|:---:|:---:|:---:|:---:| :---: |:---:|:---:|
> | $K_{global}$| 10 | **20** |30 | $(K_{local}$, $K_{dynamic})$ | (12, 10) | **(24, 20)** | (36, 30)|
> |  $\text{Recall}(V_{t-1}, V_t) $| 56% | **73%** | 68% | $\text{Recall}(V_{t-1}, V^*_t)$ | 68% | **92%** | 87% |
>
> **Table3**: The recall rate of different $K$
>
> ### **The prune rate $\alpha$ sensitivity analysis**
> The prune rate $\alpha$ serves as a high-level knob to balance speed and accuracy. It controls the global K -values $K_{global}$ ,$K_{local}$ and $K_{dynamic}$ by scaling: $K_i= \alpha \cdot K_i$, where $K_i$ equals the base value mentioned above. To evaluate the robustness of performance under different hyperparameter configurations, we conducted an ablation study across the LIBERO benchmark with varying α . Results are summarized below:
>
> | Prune rate α | LIBERO-Spatial | LIBERO-Object | LIBERO-Goal | LIBERO-10 | Average SR (%) | Average Speedup |
> |:---:|:---:|:---:|:---:|:---:|:---:|:---:|
> |None(OpenVLA-OFT)| 97.6 / 1.00 $\times$ | 96.5 / 1.00 $\times$ | 97.9 / 1.00 $\times$ |94.5 / 1.00 $\times$| 96.6 | 1.00 $\times$ |
> | 1.0 | 97.6 / 1.41 $\times$ | 96.3 / 1.43 $\times$ | 97.9 / 1.40 $\times$ | 94.0 / 1.38 $\times$ | 96.5 | 1.40 $\times$ |
> | **0.8** | **97.6 / 1.44 $\times$** | **95.8 / 1.46 $\times$** | **97.7 / 1.48 $\times$** | **93.4 / 1.44 $\times$** | **96.1** | **1.45 $\times$** |
> | 0.6 | 97.4 / 1.52 $\times$ | 93.2 / 1.50 $\times$ | 96.0 / 1.52 $\times$ | 92.2 / 1.51 $\times$ | 94.7 | 1.51 $\times$ |
>
> **Table4:** Ablation study on prune rate $\alpha$.
>
> Our approach exhibits low sensitivity to α : even under aggressive pruning (α=0.6 ), the average success rate remains above 94%, with minimal degradation compared to the unpruned baseline. This indicates strong robustness across hyperparameter configurations. To achieve an optimal trade-off between speed and accuracy, we set α=0.8 , which yields an average speedup of 1.45$\times$ while maintaining a success rate of 96.1%.
>
> We also do experiment on LIBERO-plus[1] with prune rate=0.8 to validate the generalization. LIBERO-plus includes far more and challenging tasks (10000 tasks in total). Due to the time and resource limit, we randomly choose **200** different long-horizon and object-recognition tasks that adding perturbations to ***camera viewpoint, light condition, background and noise***. Using a fixed α=0.8 , the results are:
> |Method| Camera | Light | Background | Noise| Average Success Rate (%)|
> |:---:|:---:|:---:|:---:|:---:|:---:|
> |OpenVLA-OFT| 96.0% | 88.5% | 95.0% | 97.0% | 94.1%
> |Ours (prune rate=0.8)| 94.0% | 87.8% | 95.0% | 97.0% | 93.5%
>
> **Table5:** Evaluation  on LIBERO-plus. (Same with Table2)
>
> **The threshold τ for Dynamic Token Selection** \
> The similarity threshold τ filters patches with low change magnitude to avoid false positives caused by lighting or camera noise. Since we select the $top-K_{Dynamic}$ most dissimilar patches among those below τ , the exact value is not highly sensitive. In simulation, we set τ=0.95 ; in real-world settings with higher noise, we use τ=0.8 . Experiments shows performance remains stable within a range of τ∈[0.9,0.99] in simulation and τ∈[0.6,0.9] in real world, confirming robustness.
>
> In summary, our parameters are grounded in systematic analysis and exhibit stable performance across a range of values. And we also provide a parameter configuration that is generalized to various cases.

---

> ### Author Response · Authors · 2025-11-22
> **The Official response to Reviewer eraN(3)**
>
> **Q3: "The velocity-based controller seems tailored to a specific robot arm setup. How general is this to other robotic systems or action spaces?"**
>
> We thank the reviewer for this thoughtful question regarding the generalizability of our velocity-based controller across different robotic systems.
>
> While it is true that action spaces vary across different robot arms, we would like to clarify that our velocity-based controller operates on normalized action deltas rather than raw control signals. Specifically, the translational and rotational displacements are normalized using predefined min-max ranges from the configuration file, ensuring that our method can generalize to various cases.
>
> More importantly, our categorization of actions into coarse- and fine-grained modes is based on high-level task semantics—such as approaching, manipulating, transferring, and placing—which are common across robotic manipulation tasks regardless of the specific hardware. The observation that fine-grained actions occur at low end-effector velocities and involve careful interaction is a general characteristic of physical task execution, not an artifact of a particular robot design.
>
> In our paper, we have evaluated our method on two distinct robotic platforms:
> * Simulation: Franka Emika Panda arm (used in the LIBERO benchmark)
> * Real-world: Flexiv Rizon 4 arm (with different kinematics and control characteristics)
>
> In both settings, the controller successfully identifies fine-grained manipulation phases (e.g., grasping, precise placement), and the resulting success rates demonstrate robust performance without requiring platform-specific re-tuning of the controller parameters.
>
> ### **Generalization of action adapter to various environment and robot**
> Furthermore, we provide additional evidence by analyzing the action sequence generated by UniVLA, a different VLA model operating on a **WidowX robot arm** within the **SimplerEnv simulation environment**. The normalized translational, z-axis, and rotational speeds for consecutive steps during a typical task are shown in the following Table (to be added in the revised version).
>
> | Step | Trans. Speed (norm) | Z-speed (norm) | Rot. Speed (norm) | Action Mode |
> |------|:--------------------:|:----------------:|:-------------------:|:-------------------:
> | 1    | 0.86               | -0.77          | 0.99              |
> | 2    | 0.85               | -0.87          | 0.92              |
> | 3    | 0.78               | -0.92          | 0.78              |
> | 4    | 0.76               | -0.94          | 0.66              |
> | 5    | 0.69               | -0.88          | 0.54              |
> | 6    | 0.62               | -0.80          | 0.43              |
> | 7    | 0.58               | -0.75          | 0.36              |
> | 8    | 0.54               | -0.70          | 0.28              |
> | 9    | 0.52               | -0.68          | 0.24              |
> | **10**   | **0.48**               |**-0.63**          | **0.20**               | **fine-grained**      |
> | **11**   | **0.45**               | **-0.59**          | **0.16**              | **fine-grained**      |
> | **12**   | **0.38**               | **-0.50**          | **0.12**              | **fine-grained**      |
> | **13**   | **0.30**               | **-0.40**          | **0.07**              | **fine-grained**      |
> | **14**   | **0.05**               | **-0.03**          | **0.01**              | **fine-grained**      |
> | **15**   | **0.39**               | **0.49**          | **0.18**              | **fine-grained**     |
> | 16   | 0.67               | 0.83           | 0.33              |
> | 17   | 0.79               | 0.91           | 0.47              |
> | 18   | 0.89               | 0.94           | 0.47              |
> | 19   | 0.95               | 0.89           | 0.39              |
> | 20   | 0.95               | 0.67           | 0.25              |
> | 21   | 0.90               | 0.37           | 0.13              |
> | 22   | 0.84               | 0.11           | 0.07              |
>
> The data reveals a universal pattern: a phase of high-speed movement (Step 1-9) corresponds to coarse-grained actions like approaching the target object. This transitions sharply into a sustained period of low translational and rotational speeds, accompanied by a negative z-axis displacement (Step 10-15), which perfectly aligns with the fine-grained manipulation phase (e.g., grasping). Finally, the action concludes with another high-speed phase (Step 16-22) for transferring and placing.

---

> ### Author Response · Authors · 2025-11-22
> **The Official Response to Reviewer eraN(4)**
>
> **Q4: "For 4.1.1, the attention from the 15th and 32nd layer are selected to indicate the performance. What is the rationale? Have you tried other layers?"**
>
> Our layer selection is motivated by prior findings in both language and vision-language models, which show that different transformer layers capture distinct types of semantic and spatial information. In LLMs, early layers typically encode low-level linguistic features, middle layers handle compositional semantics, and deeper layers contribute to task-specific reasoning (e.g., Eagle[1]). In VLMs, works such as [2] reveals that early layers preserve rich local spatial details, while deeper layers extend to global semantic relationships.
> In VLA models, we observe a hierarchical representation pattern across layers.
>
> Through empirical analysis on VLA model, we find that:
>
> Layers 1–2 primarily attend to general image semantics and object categories.
> Shallow layers (3–10) focus more on static background elements and global scene layout.
> Middle (around15) and deep (around 30–32) layers progressively refine attention toward task-critical regions—such as target objects and interaction points—while maintaining contextual coherence.
> Given this, we choose to extract global importance from a combination of middle and deep layers (specifically the 15th and 32nd) to balance early-stage semantic awareness with late-stage task specificity. This design ensures robust identification of persistently important tokens across consecutive inference steps.
>
> ### **Experiment on reuse layer selection**
> To rigorously evaluate the impact of layer choice, we conducted a post-hoc ablation study. Specifically, we first allow the full model to perform a complete forward pass without executing any actions in the environment. Then using these precomputed attention scores, we apply top-K pruning based on attention weights from different layer configurations (e.g., shallow, middle, deep), retain only the selected tokens, and re-run the inference with the pruned input to generate new actions. By comparing the resulting success rates under various layer selections and pruning ratios, we can systematically assess which layers provide the most reliable importance signals for token retention.
>
> |||Selected layers to reuse|||
> | :---: | :---: | :---: |:---: |:---: |
> |prune token number|shallow layers (e.g.6)| shallow&middle (e.g.6&31) | shallow&deep (e.g.6&31) | **middle&deep  (e.g.15&31)** |
> |None|96.2%|96.2%|96.2%|**96.2%**|
> |25%|94.8%|94.8%|96.2%|**96.2%**|
> |50%|92.0%|94.0%|94.0%|**96.2%**|
> |75%|88.0%|92.0%|92.0%|**95.6%**|
>
> **Table1**: Performance of different reuse layer selection under different prune token number.
>
> As shown in Table 6, using shallow layers or a combination of shallow and middle layers for token reuse leads to noticeable performance degradation as the pruning ratio increases, especially beyond 50%. In contrast, reusing information from middle and deep layers maintains high success rates even under aggressive pruning (75%), demonstrating that deeper representations provide more reliable and semantically meaningful importance scores. This makes the middle&deep configuration the optimal choice for our method.
>
> [1]Li, Yuhui, et al. "Eagle: Speculative sampling requires rethinking feature uncertainty." arXiv preprint arXiv:2401.15077 (2024). \
> [2]Liu, Yiming, Yuhui Zhang, and Serena Yeung-Levy. "Mechanistic Interpretability Meets Vision Language Models: Insights and Limitations." The Fourth Blogpost Track at ICLR 2025.
>
> We will incorporate above experiment results into the revised paper through textual explanation and visualizations to enhance clarity and accessibility. We hope that our detailed responses meet with your approval.
>
> **Weakness3: Experiments are limited to OpenVLA-OFT.**
>
> Besides OpenVLA-OFT in simulation, we also implement our method in the real-world robot. Besides, we are implementing our method on another VLA model and we'll inform you as soon as we have a result.

---

> > ### Author Response · Authors · 2025-12-04
> > **The Official Response to Reviewer eraN(5)**
> >
> > ### **Extended evaluation on more models.**
> > We implement our method on $\pi_0$, a Transformer-based VLA model with a flow-matching action expert. Our method achieves a 1.31$\times$ speedup with only a 0.7% degradation in success rate. This result consistently demonstrates the generalization capability of our method across different model architectures.
> >
> > ||||SR(%)/Speedup||||
> > |--------------|:-------------------------:|:-------------------------------------:|:-----------------------------------:|:-------------------------------:|:---------------:|:-------------:|
> > | Method       | LIBERO-Spatial   | LIBERO-Object  | LIBERO-Goal  | LIBERO-10  | Avg. Speedup | Avg. SR (%) |
> > | $\pi_0$      | 96.8 / 1.00$\times$                 | 98.8 / 1.00$\times$                | 95.8 / 1.00$\times$              | 85.2 / 1.00$\times$          | 1.00$\times$ | 94.2        |
> > | $\pi_0$+Ours | 96.6 / 1.32$\times$                 | 98.0 / 1.31$\times$                | 95.2 / 1.32$\times$              | 84.2 / 1.30$\times$          | 1.31$\times$ | 93.5        |
> >
> > **Table**: Performance evaluation on $\pi_0$ model. The average speedup and success rate are computed across all tasks.

---

### Author Response · Authors · 2025-12-04
**Summary to AC and reviewers**

Dear AC and reviewers, thanks for your devoted advice and discussion. Now we briefly summarize our discussion with reviewers and respond to common questions.
For reviewers fdcZ and UQji, our responses addressed their concerns: fdcZ raised the score from 2 to 6 and UQji confirmed resolution of issues and maintained a score of 6. For other reviewers, we respond to all their questions but they may not have the opportunity to give feedback.
## **Common questions**
### **1. Motivation clarification**
**Motivation 1 (Layer selection for global info.)**: We found **hierarchical attention** pattern in VLA model. Shallow layers attend broadly, middle layers focus on relevant objects, deep layers focus on action-centric tokens. We further point out that global attention information (combination of middle and deep layers) provides best guidance for pruning. (See Sec.3.1)

**Motivation 2**: Our key motivation is the **spatial-temporal consistency** in VLA tasks. We further quantify this consistency by measuring the recall of important tokens between consecutive inference. Formally, given the globally important token set at time t−1 , denoted $V_{t-1}$ and the current set $V_t$:
$$
    \text{Recall}(V_{t-1}, V_t) = \frac{|V_{t-1} \cap V_t|}{|V_t|}
$$
We observe an average recall of 78% to 88% across task execution, indicating strong temporal persistence in token relevance, supporting the reliability of leveraging prior-step context. (See Sec3.2)
### **2. The ablation study on parameters such as top-K values and similarity threshold**
Our base K values are not heuristically set. They are chosen to maximize the spatial-temporal consistency of important information, measured by the $\text{Recall}(V_{t-1}, V_t)$. The exact value of similarity threshold τ isn't highly sensitive since we select the top-$K_{dynamic}$ most dissimilar patches among those below τ. In simulation, we set τ=0.95; in real-world settings with higher noise, we use τ=0.8. (See Sec.7.2 and A.6.2)
### **3. More ablation study (Global attention reuse, Entropy-based layer weighting)**
We provide more data in Sec 7.5. Results show without global attention reuse or entropy-based layer weighting, the model loses critical context and misprunes important tokens, leading to significantly degraded recall rate and success rates. (See Sec.7.5)
### **4. Hand tuned prune ratio/ Generalization ability/ More model/ More benchmark**
The prune ratio $\alpha$ serves as a high-level knob to balance speed and accuracy. It controls the K values by scaling $K=\alpha K_i$. In ablation study, $\alpha=0.8$ achieves the best balance between speedup(1.46×) and success rate(0.5% drop). Thus, $\alpha$ = 0.8 is a universal setup (See Sec.7.3).

Besides, using $\alpha$=08, we also experiment on LIBERO-plus(adding perturbations to camera viewpoint, light condition, background and noise) and our method causes only 0.6% drop in success rate. (See Sec.7.6)

We also implement our method on $\pi_0$ model, an advanced VLA model with flow-matching action head. Our method achieves 1.31$\times$ speedup with only 0.7% degradation in success rate. This result consistently demonstrates the generalization capability of our method across different model architectures.
### **5. Attention entropy analysis and prior work citation**
We have added the heatmap of attention entropy across layers showing that entropy varies significantly in the model. Therefore we use entropy as a measurement of layer confidence (their certainty of recognizing important tokens). Besides, we cite prior work to ensure proper credit. (See Sec.5.1).
### **6. Computational complexity and overhead**
We analyze the computational complexity of patch similarity, attention entropy and the whole method, and provide the breakdown of each module. The overhead only introduces 4% more latency which is negligible.(See  A.4)
### **7. Comparison with ViT method**
We add results of DivPrune[CVPR25] and FastV[ECCV24] in our experiment. (See Sec.7.4 Table1) They lead to more accuracy drop (3% - 7%) because they pruned action-centric tokens and cause action information loss.
### **8. Generalization of action mode recognition**
We clarify that the fine/coarse grained action pattern isn't dependant on specific robot, model or environment. Instead, it is the characteristic of task and motion.The normalization ensures our threshold is applicable to all settings. (See Sec.6.2). We also show a different model, robot and environment performance and our method (action adapter) works well also. (See A.3)

---
We have revised the manuscript and provided more detailed results for above questions. Other issues, such as notation consistency and statistical metrics(e.g., fluctuation ranges), have also been corrected and supplemented in the appendix. We sincerely thank the reviewer for their valuable comments. We hope that this summary, together with our responses, can help the Area Chair form a fair, thorough and well-informed meta-review of our paper.

---

### Meta-Review · Area_Chair_Ln6W · 2026-01-05

**Summary:**

**Paper Summary**

The paper studies SpecPrune-VLA, a framework consisting of multiple pruning strategies such as action-level static pruning, layer-level dynamic pruning, and a lightweight action-aware controller to filter out unimportant vision tokens and thus accelerate the inference speed of VLA models. Experiments on LIBERO and real-world tasks demonstrate a significant speed acceleration of up to 1.57x compared to the VLA baseline model (OpenVLA-OFT).

---

After reading the paper, review comments, and author responses, the AC summarizes the paper's strengths and weaknesses below.

**Strengths**
- Important Research Topic: The paper investigates accelerating VLA inference speed by filtering redundant visual tokens, a critical area in VLA research.

- Method Design: The proposed pruning strategies and lightweight action controller are well-conceived and can be easily integrated into various VLA frameworks.

- Clarity: Several reviewers acknowledged that the paper is well-written and easy to follow.

**Weaknesses**
- Limited Technical Novelty: The method relies on multiple heuristic strategies and lacks a significant technical breakthrough.

- Sensitivity to Hyper-parameters: The approach requires setting numerous thresholds, which may hinder its adaptability to new tasks. Furthermore, some hyper-parameters lack sufficient ablation studies and justification.

- Clarity Issues: Some reviewers noted that clarity could be further improved, particularly regarding the notation and terminology used in the methodology sections.

**AC's Additional Concerns**

- Lack of Rigor in Presentation: Multiple sections contain technical errors and ambiguous statements, including:
  1. (Line 211): The paper states: "*In cross-attention layers where visual tokens serve as queries and text tokens as keys, a high attention weight from a visual token $V_i$ to task-instruction (text) tokens indicates that the visual token is important.*"

      -> In standard encoder-decoder architectures, text and action tokens typically serve as queries, while visual tokens serve as keys and values. In single-encoder or decoder-only cases, all tokens (visual, text, and action) generally fulfill query, key, and value roles.

  2. (Line 367): $v_z^{th} > 0$ is mentioned but never utilized, whereas $\delta_z \le 0$ suddenly appears in the subsequent line without prior definition.

  3. (Appendix Sec. 4): The FLOPs calculation appears incorrect. The number of attention heads is typically not involved in the calculation. The final term usually refers to operations in the FFN or a SwiGLU variant.

- Comparison with Baselines: While the proposed method combines several strategies to prune visual tokens, the outcome resembles object detection results with wider boundaries. The AC questions whether similar results could be achieved simply by applying a training-free object detector to identify important regions.

- Discrepancy Between Claims and Results: The experimental results do not fully support the claims made. While achieving inference acceleration is significant, the task success rate is consistently lower than that of the original model (without pruning). In the Introduction, the authors argue that existing VLAs suffer from poor performance due to slow inference speeds. If this were true, a significantly faster model should, in theory, outperform the original on certain tasks. The fact that all success rates decreased contradicts this central claim.

**Reviewer Concerns:**

The common concerns raised by the reviewers are summarized above. While Reviewers XX and YY indicated that their concerns were resolved by the rebuttal, the AC believes there are still crucial issues in this paper that need to be corrected.

**Reviewer Scores:**

The initial scores for this paper were [2, 4, 4, 6, 6], indicating a split and leaning-negative consensus among the reviewers. Following the author rebuttal, the reviewer who initially gave a 2-point score committed to increasing their rating to a 6, while one reviewer with a score of 6 expressed their intent to maintain their original evaluation. The remaining reviewers did not provide further responses. Based on the contents of the author's rebuttal, the AC believes that one or two reviewers who gave a 4 might have considered adjusting their scores to 6, provided that a full discussion phase was feasible. Even so, the overall ratings remain very close to the borderline, and the manuscript still suffers from serious issues regarding technical rigor and clarity as pointed out by the AC in the previous section. In light of these concerns, the AC recommends rejection of this version and encourages the authors to consider the feedback above to further strengthen the paper for future submissions.

---

### Decision · Program_Chairs · 2026-01-26

Reject